# Hand2 delineates mesothelium progenitors and is reactivated in mesothelioma

Karin D. Prummel[1,2,13], Helena L. Crowell [2,3], Susan Nieuwenhuize[1,2], Eline C. Brombacher[2,14], Stephan Daetwyler[4,15,16], Charlotte Soneson[2,3], Jelena Kresoja-Rakic[1,5], Agnese Kocere[1,2], Manuel Ronner[5], Alexander Ernst[6], Zahra Labbaf[7], David E. Clouthier [8], Anthony B. Firulli[9], Héctor Sánchez-Iranzo [10,17], Sundar R. Naganathan [11], Rebecca O'Rourke [1], Erez Raz [7], Nadia Mercader [6,10], Alexa Burger[1], Emanuela Felley-Bosco [5], Jan Huisken [4,12], Mark D. Robinson [2,3] & Christian Mosimann [1✉]

The mesothelium lines body cavities and surrounds internal organs, widely contributing to homeostasis and regeneration. Mesothelium disruptions cause visceral anomalies and mesothelioma tumors. Nonetheless, the embryonic emergence of mesothelia remains incompletely understood. Here, we track mesothelial origins in the lateral plate mesoderm (LPM) using zebrafish. Single-cell transcriptomics uncovers a post-gastrulation gene expression signature centered on *hand2* in distinct LPM progenitor cells. We map mesothelial progenitors to lateral-most, *hand2*-expressing LPM and confirm conservation in mouse. Time-lapse imaging of zebrafish *hand2* reporter embryos captures mesothelium formation including pericardium, visceral, and parietal peritoneum. We find primordial germ cells migrate with the forming mesothelium as ventral migration boundary. Functionally, *hand2* loss disrupts mesothelium formation with reduced progenitor cells and perturbed migration. In mouse and human mesothelioma, we document expression of LPM-associated transcription factors including Hand2, suggesting re-initiation of a developmental program. Our data connects mesothelium development to Hand2, expanding our understanding of mesothelial pathologies.

---

[1] Department of Pediatrics, Section of Developmental Biology, University of Colorado School of Medicine, Anschutz Medical Campus, Aurora, CO, USA.
[2] Department of Molecular Life Sciences, University of Zurich, Zürich, Switzerland. [3] SIB Swiss Institute of Bioinformatics, University of Zurich, Zürich, Switzerland. [4] Max Planck Institute of Molecular Cell Biology and Genetics, Dresden, Germany. [5] Laboratory of Molecular Oncology, Department of Thoracic Surgery, University Hospital Zurich, Zürich, Switzerland. [6] Institute of Anatomy, University of Bern, Bern, Switzerland. [7] Institute for Cell Biology, ZMBE, Muenster, Germany. [8] Department of Craniofacial Biology, University of Colorado Anschutz Medical Campus, Aurora, CO, USA. [9] Herman B Wells Center for Pediatric Research, Departments of Pediatrics, Anatomy and Medical and Molecular Genetics, Indiana Medical School, Indianapolis, IN, USA. [10] Centro Nacional de Investigaciones Cardiovasculares (CNIC-ISCIII), Madrid, Spain. [11] Institute of Bioengineering, Ecole Polytechnique Federale de Lausanne, Lausanne, Switzerland. [12] Morgridge Institute for Research, Madison, WI, USA. [13] Present address: Structural and Computational Biology Unit, EMBL, Heidelberg, Germany. [14] Present address: Department of Parasitology, Leiden University Medical Center, Leiden, The Netherlands. [15] Present address: Lyda Hill Department of Bioinformatics, UT Southwestern Medical Center, Dallas, TX, United States. [16] Present address: Department of Cell Biology, UT Southwestern Medical Center, Dallas, TX, United States. [17] Present address: Institute of Biological and Chemical System – Biological Information Processing (IBCS-BIP), Karlsruhe Institute of Technology (KIT), Eggenstein-Leopoldshafen, Germany. ✉email: christian.mosimann@cuanschutz.edu

As a key feature of the vertebrate body plan, the mesothelium is composed of several continuous, epithelial monolayers surrounding the internal organs (visceral mesothelium) and lining the body cavities (parietal mesothelium). The mesothelium provides a protective layer against invasive microorganisms, produces serous fluid that decreases friction of moving organs, and enables the transport of cells and nutrients across serosal cavities[1,2]. Moreover, cell tracking studies have established that the mesothelium contributes to a multitude of downstream cell fates including smooth muscles and fibroblasts during organogenesis, tissue homeostasis, and regeneration[3–5]. While mesothelium-lined body cavities are a fundamental trait across bilaterian animals[6–8], open questions remain about the lineage origins of vertebrate mesothelium.

The embryonic mesothelium, also called coelomic epithelium after establishing its baso-apical polarization, is a highly dynamic cell layer that undergoes epithelial-to-mesenchymal transition (EMT) during development and seeds mesenchymal cells to underlying tissues[4,5]. Several prior observations tie mesothelial lineage origins to the lateral plate mesoderm (LPM)[9–11], a mesodermal progenitor territory that forms at the periphery of the early vertebrate embryo[12]. In vertebrates, the coelomic cavity forms stereotypically by splitting the LPM into dorsal and ventral layers[13]. A subset of cells within both layers differentiates into polarized epithelial cells that form visceral (splanchnic) mesothelial layers and parietal (somatic) mesothelial layers[14–16]. Ultimately, the coelom in amniotes spans from the neck to the abdomen and outlines four main body compartments: two pleural cavities (around the lungs), a pericardiac cavity (around the heart), and a peritoneal (abdominal) cavity, each with their associated mesothelial layers (Fig. 1A). While teleosts such as zebrafish lack an equivalent pleural cavity due to absence of lungs, they feature mesothelium-lined cardiac and abdominal cavities. Which territories within the emerging LPM initially harbor the mesothelial progenitors, and when the mesothelium diverges from other LPM lineages, remain uncharted.

Embryonic studies of the mesothelium have predominantly focused on developmental stages after the coelomic epithelium has formed. Several genes, including *Mesothelin* (*Msln*), *Gata4*, *Tbx18*, *Tcf21*, and *Wilms Tumor 1* (*Wt1*), have enabled labeling and genetic lineage tracing of mesothelial lineages in mouse and chick[5,17–20]. The *Wt1*-expressing coelomic epithelium can contribute to the mature mesothelium, fibroblasts, stellate cells, smooth muscles, and white adipose tissue associated with the gastrointestinal tract, liver, lungs, urogenital system, and heart[5,10,19,21–27]. Further, regional specification within the mesothelial components along the developing gut is recognizable early in mouse development[28,29]. Studies using zebrafish have documented expression of *wt1a/b*, *tcf21*, and *tbx18* in the epicardium, the visceral mesothelial layer covering the heart[30,31]. Despite these advances across models, expression of these conserved genes is initiated after the onset of coelomic epithelium formation, leaving the earliest differentiation steps obscure.

Compromised integrity of the adult mesothelium can result in pathologies including intra-abdominal organ adhesion[32], serosal fibrosis[33], pericarditis[34], and mesothelioma tumors[35,36]. Malignant mesothelioma is a rapidly fatal solid tumor that can arise within the visceral or parietal mesothelia, predominantly as the result of environmental exposure to asbestos[35–39]. While mesothelioma cases are increasing globally despite regulatory means to curb the use of causative agents, treatment remains limited[35,40]. Presenting predominantly as epithelioid, sarcomatoid, and biphasic phenotypes, malignant mesothelioma frequently harbor genetic alterations affecting the tumor suppressors *BAP1*, *NF2*, *CDKN2AB*, and *TP53*[41–45]. Nonetheless, the cell of origin and the underlying aberrant molecular mechanisms leading to mesothelioma remain uncertain.

In addition to the mesothelium, the LPM gives rise to a vast array of downstream cell fates that include the cardiovascular system, blood, kidneys, and limb connective tissue[12]. How the LPM partitions into its diverse fates and what regulatory programs specify the individual progenitor fields remain unclear. Emerging as a dedicated mesendoderm domain, the post-gastrulation LPM segments into recognizable bilateral territories discernible by the expression of several transcription factor genes including *Scl/Tal1, Lmo2, Pax2a, Nkx2.5*, and *Hand1/2*[12,46–48]. *dHand/Hand2*, encoding a conserved basic helix-loop-helix transcription factor, is expressed during segmentation stages in the most laterally positioned LPM progenitors in amphioxus and zebrafish[13,49,50]. In the developing mouse embryo, *Hand2* expression has been described in the flank at comparable embryonic stages[51–53]. Studies across vertebrate models have revealed key insights into the role of Hand2 and its paralog eHand/Hand1 in anterior LPM (ALPM) progenitors that contribute to the heart, (fore)limbs, and branchial arches[54–56]. Additionally, in the posterior LPM of zebrafish and chick, Hand2 has been linked to refining the fate divergence between smooth muscle versus hemangioblast and kidney fates during somitogenesis[49,57,58]. The definitive fate of especially the posterior, lateral-most Hand2-expressing LPM progenitors has remained unclear.

Here, we establish several lines of evidence that *hand2* in zebrafish is the earliest specific transcription factor gene demarcating the emerging mesothelial progenitors within the LPM. We provide further evidence for conservation of this property in mouse. We link the developmental function of Hand2 in mesothelium formation to a reactivation of an early coelomic epithelium-focused LPM program in mouse and human mesothelioma tumors. Our findings propose that Hand2 expression contributes to the unique properties of mesothelial progenitor cells in development and in mesothelioma.

## Results

**Zebrafish mesothelium is LPM-derived**. To formally assess whether in zebrafish mesothelial membranes are bona fide LPM lineages, we performed genetic lineage tracing using *drl:creERT2*. From the onset of gastrulation until early-to-mid somitogenesis, *drl:creERT2* is active in LPM-primed mesendoderm cells, resulting in lineage labeling of seemingly all LPM-assigned mesodermal lineages including cardiac, endothelial, hematopoietic, kidney, intestinal smooth muscles, and craniofacial muscles, with progressively waning labeling of the endoderm over time[50,58–64] (Fig. 1A, B). We induced *drl:creERT2;hsp70l:Switch* embryos with 4-OH-Tamoxifen (4-OHT) at shield stage and analyzed the resulting EGFP-based LPM lineage labeling in transverse sections at 3 days post-fertilization (dpf) when LPM-derived organs are clearly detectable (Fig. 1A, B). EGFP expression recapitulated broad labeling of LPM-derived organs including endothelium, blood, and cardiac lineages, and sparse labeling of endoderm-derived organs, in line with previous observations using *drl:creERT2*[50,59] (Fig. 1C–G and Supplementary Fig. 1A–C). Extending the previously reported LPM lineage labeling in the Transgelin (Tagln)-positive smooth muscle layers around the zebrafish gut (Fig. 1C)[58], we also observed smooth muscles around the swim bladder and ducts within the liver by double-positive staining for EGFP and Tagln (Fig. 1C). In addition, we consistently observed EGFP-positive, yet Tagln-negative cells as thin epithelial layers surrounding the gut and swim bladder (Fig. 1C), as well as around other endodermal organs including the liver and the pancreas (Fig. 1D, E). We also found

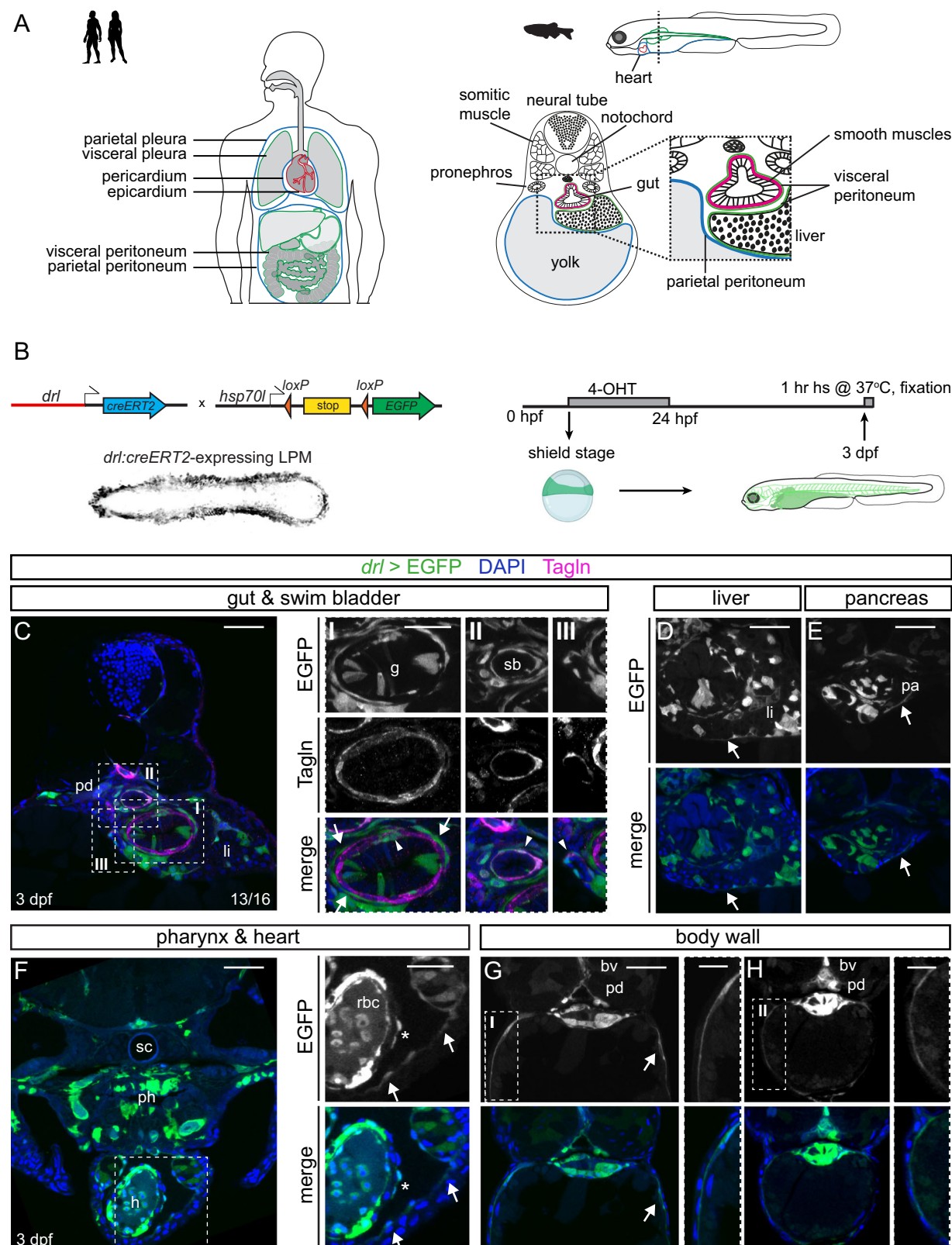

EGFP-lineage labeling in the pericardial and epicardial layers surrounding the heart, confirming their LPM origin (Fig. 1F and Supplementary Fig. 1A, D–I). Of note, at this stage, the pericardium directly adheres to the body wall, forming a sac around the cardiac cavity, while in adult zebrafish the pericardium establishes a dedicated body cavity. Lastly, we observed prominent *drl*-based LPM lineage labeling within the body wall as an epithelial layer right underneath the skin surrounding the yolk and the yolk extension along the anterior–posterior axis (Fig. 1G, H). From these observations, we conclude that the prospective coelomic epithelium and most, if not all, developing mesothelial layers in zebrafish are LPM-derived and that *drl*-based reporters are suitable to label early mesothelial progenitors as part of their pan-LPM activity.

**Fig. 1 Visceral and parietal mesothelial layers in zebrafish are LPM lineages. A** Mesothelium in human versus zebrafish embryo. Transverse schematic of zebrafish embryo showing liver, gut, and budding swim bladder with associated mesothelium and smooth muscle at 3 dpf. **B** Tracing LPM using *drl:creERT2* × *hsp70l:Switch*, 4-OHT administered at shield stage and washed off before 24 hpf. *drl>EGFP* indicates LPM lineage labeling. **C** Trunk section of *drl* lineage-traced 3 dpf embryo co-stained for smooth muscle (Tagln). Boxed regions show details of EGFP lineage-labeling and smooth muscle-labeling around gut (box I), swim bladder (box II), and liver ducts (box III). Arrows depict Tagln-negative;EGFP-positive cells, arrowheads depict Tagln;EGFP double-positive cells. **D, E** EGFP-based LPM labeling in the peritoneum around liver (**D**) and pancreas (**E**). **F** Rostral transverse section, lineage labeling of pericardium, ventricle (endocardium, myocardium, potentially epicardium (asterisk), blood), and LPM-derived and endoderm-derived organs in the head (head cartilage, vasculature, pharynx). **G, H** Sections of two regions along the anterior-posterior axis, **G** at yolk, **H** at yolk extension, showing *drl*-based LPM lineage labeling of parietal peritoneum forming body wall together with skin layer (boxed region). Pronephric duct (pd), liver (li), gut (g), swim bladder (sb), pancreas (pa), heart (h), blood vessel (bv), spinal cord (sc), pharynx (ph), and red blood cells (rbc). Nuclei in blue (DAPI). Scale bars **C–H** 50 μm and 25 μm (boxed regions).

**The early LPM harbors distinct *hand2*-expressing cells**. We next sought to chart the mesothelial progenitors within the emerging LPM. To probe whether early mesothelial progenitors can be recognized within the emerging LPM, we analyzed the transcriptome of individual zebrafish LPM cells by single-cell RNA-sequencing (scRNA-seq) at tailbud stage (Fig. 2A). We dissociated a pool of *drl:mCherry*-expressing zebrafish embryos and sorted the mCherry-expressing cells. At this stage, the end of gastrulation, the *drl:mCherry* reporter-labeled LPM comprises approximately 7% of all cells in the zebrafish embryo (Fig. 2B and Supplementary Fig. 2). Upon quantifying the transcriptomes from individually sorted *drl:mCherry*-positive cells using CEL-Seq2[65], we obtained 1039 cells that passed filtering and quality control. Using graph-based clustering (Louvain algorithm[66]), we called 15 distinct cell populations within the *drl*-positive LPM at tailbud stage (Fig. 2C). We annotated these as contributing to eight major subpopulations based on canonical markers and published gene expression patterns in conjunction with marker genes identified through differential expression analysis (Fig. 2E, see "Methods" section for details). Of note, while we collected a carefully stage-matched pool of embryos, the timeframe of zebrafish matings and embryo collection might lead to slight heterogeneity among developmental stages; consequently, instead of representing bona fide distinct clusters, individual related clusters could potentially represent different developmental times of the same precursors (i.e. hemangioblasts and kidney_1 and _2).

While mRNA in situ hybridization (ISH) and transgene expression for the earliest markers of individual LPM fate potentials renders them chiefly detectable from early somitogenesis on[48], our analysis at tailbud stage resolved seemingly determined LPM progenitor fields already at the end of gastrulation, in line with and extending previous findings in zebrafish and mouse embryos[59,67–71]. Our uncovered clusters broadly represent the cardiopharyngeal, emerging head mesoderm, hatching gland progenitors, endoderm, and endothelial and hematopoietic progenitors (Fig. 2C, E). Two clusters were composed of presumptive cardiopharyngeal progenitors based on the expression of *tbx20, nkx2.5, nkx2.7, hey2, gata4/5/6, ttn.1,* and *ttn.2* (Fig. 2C, E)[72–78]. One cluster was positive for designated markers of putative head mesoderm progenitors, including *fsta, foxc1a, gsc, meox1,* and *prrx1a*[79]. Based on the expression pattern of *alx1, pitx2,* and *pitx3,* we assigned a cluster to represent the hatching gland progenitors[80,81], in accordance with *drl* reporter activity in these cells[59]. We additionally uncovered cells expressing the endodermal genes *sox32, sox17, gata5, gata6, foxa2,* and *fox3a*[67,70,82,83] as distinct group of clusters within the analyzed cells (Fig. 2C, E). This observation is in line with previous findings that the *drl* reporter-positive cells at tailbud stage represent either mixed endoderm-primed and LPM-primed populations or a bi-potential LPM-fated mesendoderm population[50].

Towards reconstructing rudimentary positional information back to the dissociated single cells, we mapped expression of the homeobox transcription factor gene *cdx4* and several *hox* genes to assign whether a cluster likely represented anterior or posterior cells within the embryo (Fig. 2D and Supplementary Fig. 3). *cdx4* at the five somite stage broadly demarcates in the posterior half of the developing zebrafish embryo (Fig. 2D). *cdx4*-positive and thus posterior clusters composed of cells expressing marker genes for endothelial and hematopoietic precursors (*fli1a, lmo2,* and *znfl2a*)[84–86], and for the pronephros (*pax2a, pax8,* and *lbx2*)[49,87,88] (Fig. 2E). The overlapping expression of hemangioblast and kidney markers possibly indicates that these two clusters represent a mixed multi-lineage progenitor pool at the end of gastrulation.

Notably, among all the prominent LPM genes we detected *hand2* expression across several clusters. In addition to clusters encompassing the expected cardiac and cardiopharyngeal precursors, *hand2* transcripts were abundant in four clusters we accordingly named *hand2*-high_1–4 (Fig. 2C, E). Further, expression of *hand2* fell into both anterior and posterior expression domains, correlating with its native expression pattern in zebrafish[54] (Fig. 2C, E). As anticipated, *hand2* transcripts in anterior LPM cells coincided with the expression of cardiac genes in one designated cluster (Fig. 2E). However, the anterior *hand2*-high_3 and *hand2*-high_4 clusters showed no detectable expression of cardiac markers including *hey2, tbx20, nkx2.5,* and *nkx2.7*, suggesting that they encompass cells with another fate potential. We also found posterior *hand2*-positive cells that appeared distinct from endothelial, hematopoietic, and kidney progenitors (*hand2*-high_1 and *hand2*-high_2) (Fig. 2E). Taken together, our scRNA-seq captured *hand2* expression as a central feature of several LPM progenitor clusters.

We next aimed to determine what genes are co-expressed with *hand2*. Our analysis revealed that several genes were enriched and individually even among the cluster-defining genes in one or more of the *hand2_high* clusters, including *sfrp5, foxh1, gata5, gata6,* and *meis3* (Fig. 2F–K and Supplementary Fig. 4). In fluorescent mRNA ISH (RNAscope) and colorimetric mRNA ISH on whole-mount tailbud and early somitogenesis staged embryos, we observed that the endogenous expression pattern of *meis3* expression overlaps with the *hand2* domain in the anterior and posterior LPM, along with *meis3* expression in other non-LPM domains (Fig. 2G, I). Moreover, *foxh1* expression has previously been described in the LPM during somitogenesis stages[89,90], yet the exact domain within the LPM has remained unclear. Akin to *meis3*, our fluorescent ISH confirmed that *foxh1* is expressed in the lateral-most posterior LPM territory overlapping with *hand2* expression (Fig. 2H). Notably, the in part redundant transcription factor genes *gata4, gata5,* and *gata6* that play key roles in cardiac and endoderm development[91], are also expressed in a domain lateral to the forming heart field[92,93]. The cells in this domain take on a spread-out, mesh-like pattern over the yolk during somitogenesis (Fig. 2J). Similarly, we found that *sfrp5* demarcates lateral-most LPM cells that form a spread-out expression domain along the body axis over the yolk and yolk extension (Fig. 2K).

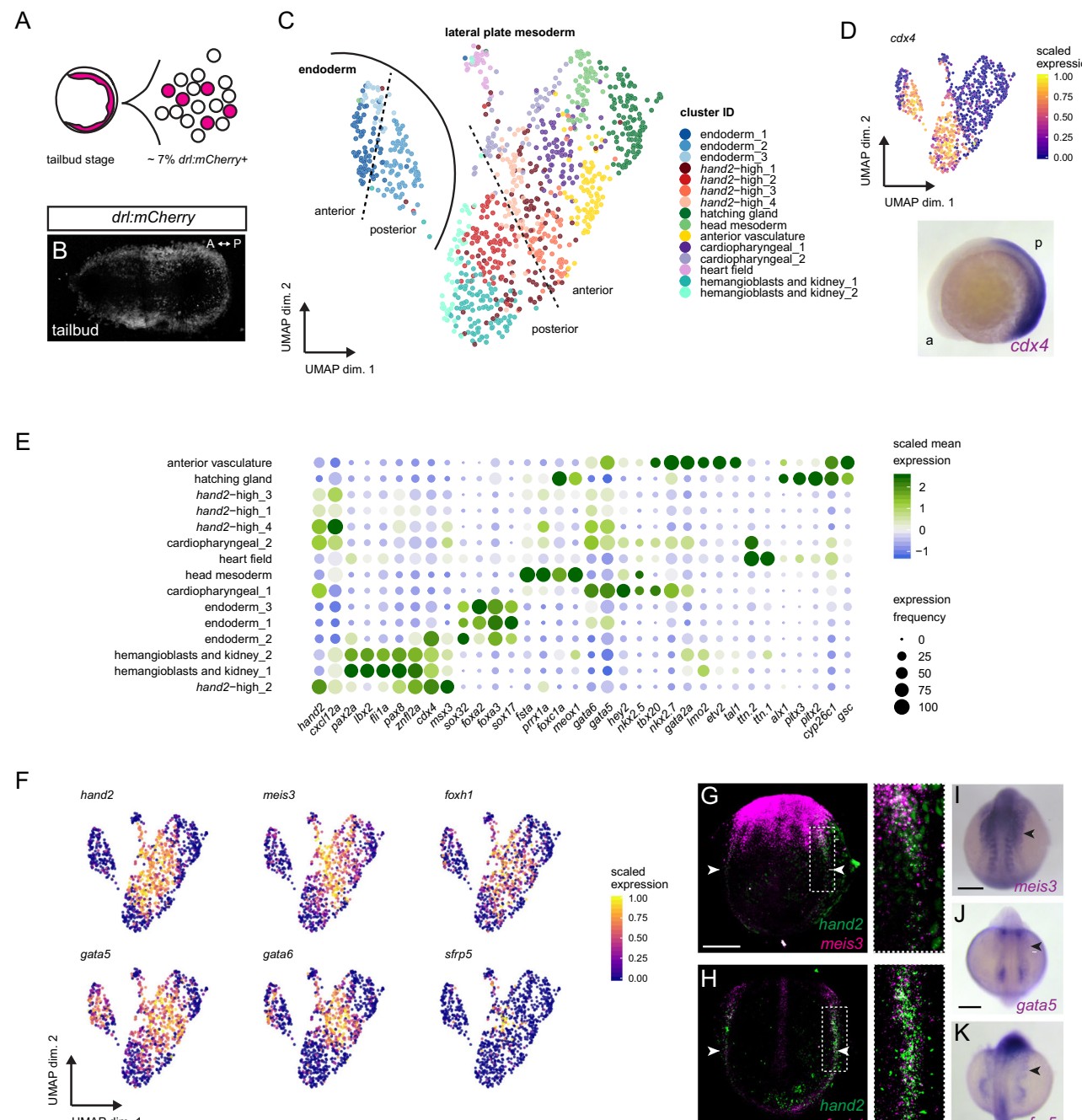

**Fig. 2 scRNA-sequencing of early LPM reveals a heterogeneous progenitor pool. A** LPM-marking *drl:mCherry*-positive cells of zebrafish embryos at tailbud stage, FACS-isolated, and sequenced using CEL-Seq2. **B** SPIM projection of *drl:mCherry* embryo at tailbud, labeling the LPM. **C** UMAP plot displaying 15 LPM cell clusters, colored by subpopulation: cardiopharyngeal (1,2), heart field, anterior vasculature, hemangioblasts/kidney (1,2), head mesoderm, hatching gland, *hand2*-high (1–4), and endoderm (1–3) as distinct group. **D** Approximate anterior-to-posterior orientation based on *cdx4* (RNA ISH at 5 ss). **E** Dotplot including key cell fate markers to annotate clusters. Dots colored by column-scaled mean expression (log-transformed library-size-normalized counts) and sized by expression frequency (fraction of cells with non-zero counts); rows and clusters ordered by hierarchical clustering of scaled expression values. **F** UMAP plots of several genes co-expressed with *hand2* or among cluster-determining genes in four *hand2*-high clusters. Cells colored by scaled expression values using lower/upper 1%-quantile boundaries. **G**–**K** Whole-mount gene expression analysis of select transcripts enriched in *hand2*-high cells by fluorescent in situ hybridization (ISH) (**G**, **H**) and colorimetric mRNA ISH (**I**–**K**). Fluorescent ISH of *meis3* (**G**), *foxh1* (**H**) with *hand2* at 4 ss, revealing overlap in posterior LPM (magnified regions from dashed boxes). *meis3* ISH at 10 ss (**I**), *gata5* at 12 ss (**J**), and *sfrp5* at 18 ss (**K**) showing expression in lateral-most LPM sprawling outwards (arrowheads). Scale bar **G**, **H** 100 μm, **I**–**K** 250 μm.

This mesh-like pattern can also be recognized within the endogenous expression of *hand2* within the ALPM[54]. Taken together, our data documents a collection of genes co-expressed with *hand2* in the lateral-most LPM domain of as-of-yet unclear fate.

**hand2 expression identifies mesothelial progenitors.** Our analysis indicated that expression of *hand2* and several associated genes demarcate a lateral-most LPM domain that is distinct from the cardiopharyngeal, blood, vasculature, and kidney progenitors at the end of gastrulation. Based on the previous association of

Hand1/2 expression with differentiated mesenchymal structures in mouse and chick[11,49–52,94], we hypothesized that the *hand2*-expressing LPM in zebrafish, in particular the posterior lateral-most stripe, forms the mesothelium. We turned to the transgenic line *hand2:EGFP* based on a BAC encompassing the zebrafish *hand2* locus that faithfully recapitulates endogenous *hand2* expression and that has been previously described to be expressed in mesenchymal cells[55]. Using time-lapse SPIM imaging and panoramic projections, we captured the dynamics of *hand2:EGFP* reporter activity during segmentation stages, demarcating the lateral-most *drl*-expressing LPM (Fig. 3A, B)[49,50]. Notably, and in contrast to the more medial LPM stripes that progressively migrated to the midline, a subset of the *hand2*:EGFP-expressing cell population sprawled out laterally over the yolk as a single-cell layer (Fig. 3C).

In transverse sections of *hand2:EGFP*-transgenic embryos at 3 dpf, we observed the previously described *hand2* reporter expression in Tagln-positive, LPM-derived intestinal smooth muscle cells that layer around the endodermal gut tube[58] (Fig. 3D). In higher magnification analysis of transverse sections, we also observed Tagln-negative *hand2*:EGFP-expressing cells surrounding the gut at 3 dpf, reminiscent of coelomic epithelial cells (Fig. 3D). At 3 dpf, the *hand2* reporter-expressing layers had also wrapped around other endoderm-derived organs, including the liver and the pancreas (Fig. 3D, E). In addition, in transverse sections capturing the yolk at 3 dpf, we detected *hand2*:EGFP expression underneath the skin in the body wall, the prospective parietal peritoneum (Fig. 3F). Extending previous observations[49,55,58,95], these data suggest that *hand2:EGFP* reporter expression at 3 dpf in zebrafish delineates the visceral and parietal peritoneal membranes.

To extend our interpretation of zebrafish *hand2* as early marker for the emerging mesothelium, we next turned to the mouse to uncover any mesothelial lineage contribution of cells expressing either of the two partially redundant murine *Hand* genes *Hand1* and *Hand2*. Crossing either the *Hand1*[EGFPCreΔNeo/+96] or *Hand2*[Cre97] transgenic strains into the *R26R loxP* reporter strain[98] resulted in lineage labeling along the mesothelium lining of the liver lobes at E14.5 and E15.5, respectively (Fig. 3G, H). We found lineage labeling to be more apparent and widespread using the *Hand1*[EGFPCreΔNeo/+] strain, with strong staining in most of the mesothelium and in a subset of cells within the liver of currently unknown identity (Fig. 3G). Labeling was also present in some cells lining sinusoids, though more sporadic. Lineage analysis of *Hand2*[Cre]-descendant cells also revealed lineage labeling in the mesothelium (Fig. 3H), though staining was more restricted and less robust than that observed in *Hand1* daughter cells. These differences could reflect differences between the two Cre drivers (knockin vs. transgenic) or a larger proportion of *Hand1*-expressing cells contributing to the liver mesothelium.

To further investigate a possible conserved link of *Hand* gene expression to mesothelium formation in mouse, we mined recent scRNA-seq and scATAC-seq data of early mouse embryos[99] (Supplementary Fig. 5). This data harbored *Hand2*-expressing cell clusters that were previously assigned as "mesenchymal", and that share co-expression of transcription factor genes we found in our zebrafish LPM dataset as associated with mesothelial progenitors, including *Meis3*, *Gata5*, and *Gata6* as well as the previously LPM-associated genes *Gata4*, *Foxf1*, and *Hand1*[12] (Fig. 2 and Supplementary Fig. 5). Together, these observations indicate that also in mammals *Hand* gene expression is a conserved feature of the mesothelial progenitors already at early developmental stages.

To further confirm and define the *hand2*-expressing mesothelia, we turned to Wt1 expression that characterizes the developing visceral mesothelium in mammals[100–102]. Expression of the zebrafish *Wt1* paralogs *wt1a* and *wt1b* becomes detectable at

6–8 ss as previously analysed in kidney and epicardium development[30,103–105]. During somitogenesis, we found *wt1a:EGFP* and *wt1b:EGFP* activity lateral of the differentiating kidney structures within the *drl:mCherry*-expressing LPM corresponding to the *hand2*-positive territory (Supplementary Fig. 6A, B). In transverse sections of *wt1a:EGFP*-expressing and *wt1b:EGFP*-expressing embryos, we detected EGFP signal in the coelomic epithelium surrounding the gut, liver, and pancreas in addition to the previously reported labeling of the glomerulus and pronephric tubules (Fig. 3I, J)[30,103–105]. Further, we observed *wt1a/b* reporter-expressing Tagln-positive smooth muscle cells around the gut and hepatic and pancreatic ducts (Fig. 3I, J). Genetic lineage tracing using *wt1a-* and *wt1b:creERT2* from the onset of transgene expression at 6–8 ss robustly marked the visceral peritoneum around the gut, liver, and pancreas (Supplementary Fig. 6C–E). Notably, at 3 dpf, we neither detected any EGFP-labeling of the visceral peritoneum around the more posterior gut, nor did we detect any EGFP-expressing parietal peritoneum cells around the yolk, indicating these cells do not express *wt1* genes in zebrafish at our analyzed time points (Supplementary Fig. 6F–G). Extending previous *wt1a*-based lineage tracing[106]. We observed that also *wt1b*-expressing cells contribute to the cardiac mesothelial layers in the dorsal and ventral pericardium, and in the pro-epicardial clusters (Supplementary Fig. 6H, I). These observations are consistent with LPM lineage tracing using *drl:creERT2* to ventral and dorsal pericardium and to pro-epicardium (Fig. 1F and Supplementary Fig. 1D–I). We conclude that, akin to the mammalian mesothelium, expression of both *Wt1* orthologs is also a feature of developing mesothelia in zebrafish. Nonetheless, in addition to the absence of *wt1a/b* expression in the parietal peritoneum lineage, *hand2:EGFP* (and *hand2* gene expression itself) presents an earlier and more complete marker of all developing mesothelial membranes in zebrafish.

**Primordial germ cells associate with mesothelial progenitors.** We noted the chemokine-encoding gene *cxcl12a* as significantly co-expressed with *hand2* in our scRNA-seq data set (Figs. 2E and 4A). Cxcl12a provides directional cues for the migration of several cell types[107], including the guidance of primordial germ cells (PGCs)[108]. Through complex migration paths, the PGCs reach their final destination in the region where the gonad develops, dorsally and medially bordered by the developing pronephros, while the identity of the ventrally located tissue has not been precisely defined[109–111]. We asked whether the *drl-* and *hand2*-expressing mesothelium interacts with the migrating PGCs and if PGCs are possibly already associated with the developing LPM during gastrulation. In time-lapse imaging of transgenic zebrafish embryos expressing EGFP in the LPM (*drl:EGFP* transgene) and farnesylated mCherry in their PGCs (*kop:mCherry-f'-nos3'* UTR)[112], we observed that the migrating PGCs associate with the *drl* reporter-expressing mesendoderm already during gastrulation (Fig. 4B, C and Supplementary Movie 1). The *drl:EGFP*-expressing progenitor field is gradually specifying into the LPM at the beginning of somitogenesis[50] and the PGC clusters continued to associate with, and migrate within, the forming LPM (Fig. 4B, C). By 24 hpf, the PGCs reach the region where the gonads will form[113] and we observed that the *hand2*-positive mesothelium-primed LPM is associated with the PGCs throughout these stages. Imaging *mCherry-f'-nos3'UTR;hand2:EGFP* embryos documented how the PGCs migrated within the *hand2:EGFP*-expressing LPM, and the PGCs eventually arrive at a domain within the *hand2*-expressing forming mesothelium (Fig. 4D). Antibody staining for the kidney transcription factor Pax2a confirmed that the PGCs end up ventral of the pronephros within the *hand2*-positive LPM

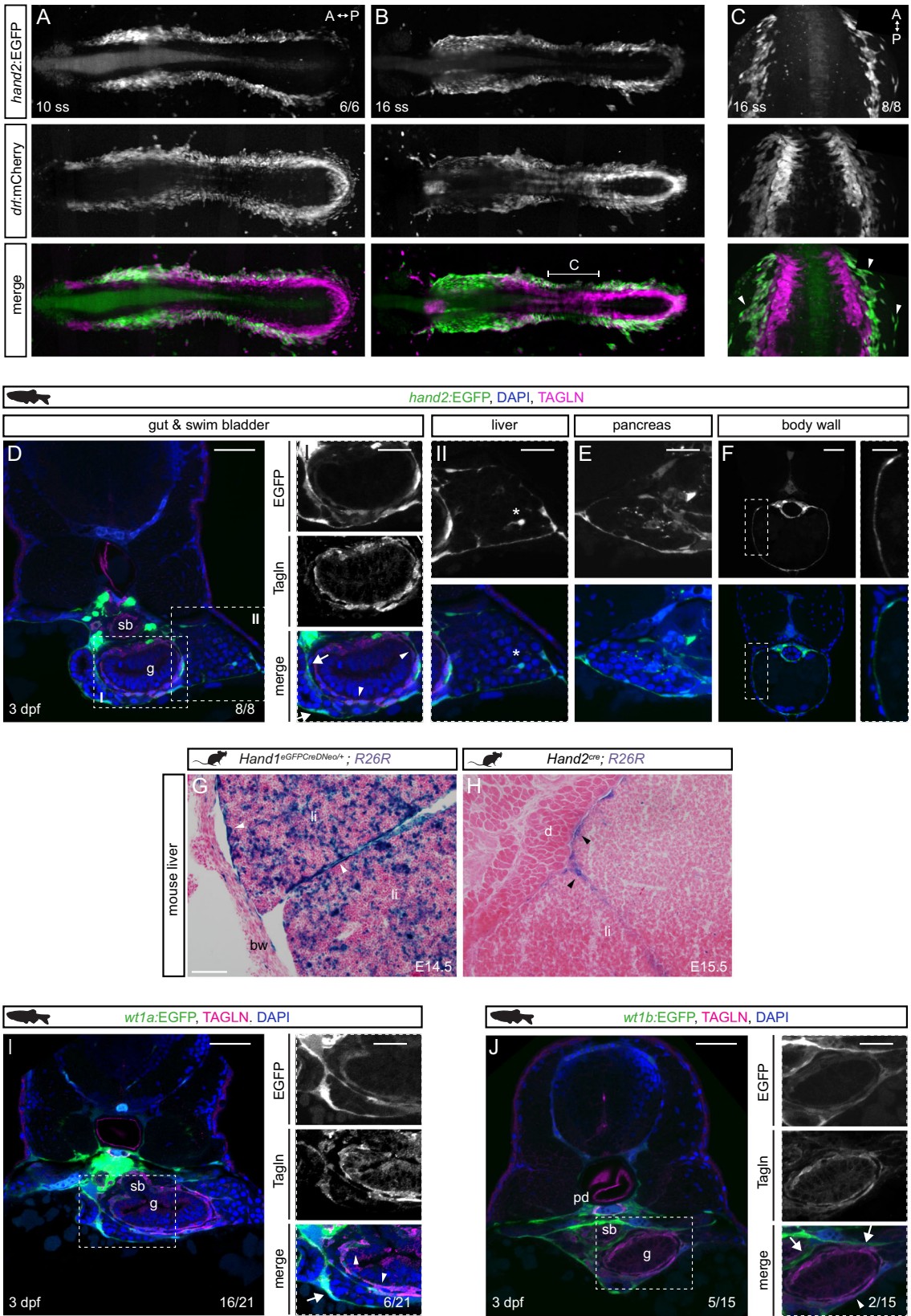

(Fig. 4E, F), with the developing gut located medially between the PGC clusters[114]. These observations further support our conclusion that the PGCs migrate in close association with the forming coelomic epithelium, as marked by *hand2* expression, already during early development. The functional importance of the *hand2*-expressing cells for the positioning of zebrafish germ cells at their target location has been previously examined[115]: while arriving at their targeted location, PGCs dispersed in embryos lacking Hand2 function. Together, the data suggest a model in which that following their initial arrival at their target

**Fig. 3 hand2 defines mesothelial precursors in the LPM. A**, **B** SPIM projections of *hand2:EGFP;drl:mCherry* embryos at 10 ss (**A**) and 16 ss (**B**). **C** Confocal imaging of 16 ss *hand2:EGFP;drl:mCherry* embryo showing *hand2:EGFP*-expressing cell populations comprising lateral-most LPM. Arrowheads label single-cell layer laterally migrating over yolk. Region as annotated in **B**. **D–F** Transverse sections of 3 dpf *hand2:EGFP* embryos. **D** Boxed regions show double-positive Tagln staining (arrowheads) and Tagln-negative staining in visceral peritoneum around (I) gut, pancreatic/hepatic ducts (arrows), and (II) liver. Asterisk depicts a potential hepatic stellate cell. **E** *hand2*:EGFP expression in visceral peritoneum surrounding the pancreas. **F** Transverse section of *hand2*:EGFP labeling of parietal peritoneum. **G**, **H** Abdominal transverse sections of E14.5 *Hand1^EGFPCreΔNeo/+;R26R* and E15.5 *Hand2^Cre;R26R* mouse embryos, lineage-labeled cells marked by β-galactosidase staining (blue). In both groups, lineage labeling appears in the visceral peritoneum of the liver (arrowheads). **I**, **J** *wt1a*:EGFP (**I**) and *wt1b:EGFP* (**J**) expression in pronephric ducts and visceral peritoneum surrounding the gut, swim bladder, and liver. Boxed regions show double-positive Tagln staining (arrowheads) and Tagln-negative staining (arrows). Body wall (bw), diaphragm (d), gut (g), liver (li), pronephric duct (pd), swim bladder (sb). Nuclei in blue (DAPI). Scale bars **D**, **F**, **I**, **J** 50 μm, **E**, boxed regions **D**, **F**, **I** 25 μm, and **G**, **H** 100 μm.

region, the PGCs are kept in place by Cxcl12a, while in *hand2* mutants the cells expressing the chemokine are affected, thereby leading to mispositioning of the germ cells.

**hand2-expressing LPM forms visceral and parietal mesothelium.** To clarify their mesothelial identity and to resolve the dynamics of *hand2:EGFP*-expressing mesothelial progenitor cells in the zebrafish embryo, we performed long-term and *in toto* light sheet imaging of developing *hand2:EGFP;drl:mCherry* double-transgenic embryos. Applying a multi-sample imaging and processing workflow[116], we captured the dynamics of the *hand2:EGFP* expression over the course of embryonic development from 18 to 82 hpf *in toto* (n = 6 embryos). At imaging onset, the *hand2*:EGFP-expressing cells had initiated their lateral migration over the yolk, while expression of *drl:mCherry* refined to cardiovascular and blood lineages (Fig. 5A and Supplementary Movie 2)[59].

In the resulting time lapse imaging, we captured the *hand2:EGFP*-positive LPM as a dynamic cell population with distinct migration behaviors in different regions of the developing embryo. First, we observed the *hand2:EGFP*-expressing cells contributing to the previously described processes of pharyngeal arch formation, pectoral fin bud outgrowth, and primary heart tube extension (Fig. 5B and Supplementary Movie 2 and 3). Additionally, our light sheet imaging registered that two seemingly separated populations of *hand2*:EGFP-expressing cells migrate ventro-laterally, crawling over the yolk and yolk extension, and continue to form the parietal layer of the cardiac cavity (pericardium) and abdominal cavity (parietal peritoneum), respectively (Fig. 5C–G). Already at 18 hpf, we observed a dorso-ventral split in the *hand2*-expressing progenitor field positioned lateral of the cardiopharyngeal progenitors, forming the cardiac cavity (Fig. 5E). The pericardium surrounds the forming heart tube and maintains the pericardial fluid that supports the beating, moving heart[31]. The cells that will form the dorsal pericardium migrated over the yolk as a connected cell field with a clearly visible leading edge, and were already interacting with the forming primary heart tube (Figs. 3B and 5D and Supplementary Movie 2 and 3). We observed that the cells forming the more posterior-located parietal peritoneum are split from the forming pericardium by the circulation valley forming over the anterior yolk, where later the common cardinal vein (Duct of Cuvier) migrates into to ultimately enclose the circulating blood[117] (as visualized by *drl*:mCherry, Supplementary Movie 1 and Supplementary Fig. 7). In addition, we documented how the *hand2*:EGFP-cells forming the presumptive parietal mesothelium reach the ventral midline earlier around the caudal yolk extension than around the main yolk, possibly due to the shorter migration distance over the yolk extension's smaller diameter (Fig. 5F–H). Simultaneously, the more medially located *hand2:EGFP*-expressing cells migrated towards the midline (Fig. 5F–H), corresponding to the formation of visceral peritoneum including intestinal smooth muscles[55,58]. Taken together, these imaging data further

support the notion that *hand2:EGFP*-expressing LPM progenitors emerging at the end of gastrulation in zebrafish form the visceral, parietal, and pericardial mesothelium.

**hand2 mutants fail to form intact mesothelia.** The prominent expression of *hand2* in mesothelial progenitors of the LPM prompted us to investigate whether the loss of *hand2* has an impact on the formation of the mesothelial layers. Phenotype analyses of mutant mice and zebrafish have documented that Hand2 (partially redundant with Hand1 in mice) contributes to the development of the heart, forelimbs/pectoral fins, and smooth muscle lineages[54,118–122]. Additionally, zebrafish *hand2* mutants feature disorganized migration of the intestinal smooth muscle progenitors towards the gut and overall gut mislooping due to perturbed extracellular matrix remodeling[55,58].

We revisited the previously established zebrafish mutant *han^S6* that harbors a presumptive *hand2* null allele[54] as well as the established *hand2* translation-blocking antisense morpholino[49,123–125]. In addition to the well-described cardiac and pectoral fin defects, we observed that *han^S6*-homozygous embryos displayed ventral defects of their yolk extensions and subsequent deterioration of the embryo due to yolk herniation at 3 dpf (Fig. 6A–F). Moreover, we noticed in *han^S6* mutants selective blistering and shriveling of the ventral fin fold (Fig. 6B, E) and uneven distribution of mesothelial progenitors as marked by *hand2:EGFP* (Fig. 6C, F and Supplementary Movie 4). At the cellular level, in contrast to the connected epithelial layer forming mesothelia in wildtype embryos, we observed that LPM-derived cells (marked by genetic lineage tracing using *drl:creERT2*) are sparse in the abdominal cavity in *han^S6*-homozygous embryos (Fig. 6G). The sparse distribution of LPM-derived cells could potentially result from mosaic labeling by incomplete *loxP*-reporter recombination following *drl*-driven CreERT2 activation, as common in Cre/*loxP*-based experiments[126]. However, *han^S6*-homozygous embryos and *hand2* morpholino-injected embryos displayed similar migration issues when we directly visualized for the mesothelial cells with the *wt1a:EGFP* and *hand2:EGFP* transgenic reporters, respectively (Fig. 6G–J and Supplementary Fig. 8). Quantification of the cells labeled by *hand2:EGFP* expression over the first 3 days of development based on *in toto* datasets of wildtype versus *hand2* knockdown further indicated a reduction and stagnation of cell number over time: while wildtype embryos steadily increase the number of *hand2*:EGFP-expressing cells, embryos with perturbed *hand2* harbor less *hand2:EGFP*-expressing cells and their numbers fail to increase at the same pace, most-dramatically over the trunk (Fig. 6G and Supplementary Fig. 8). Additionally, in *hand2* mutant *wt1a:EGFP* embryos, we observed random EGFP-expressing cells in the body wall, while parietal mesothelium cannot be detected in *wt1a:EGFP* wildtypes (Fig. 6I and Supplementary Fig. 6), further strengthening our findings of aberrant mesothelial progenitor migration. Together, our observations indicate that loss of *hand2* perturbs mesothelial progenitor migration and reduces their overall cell

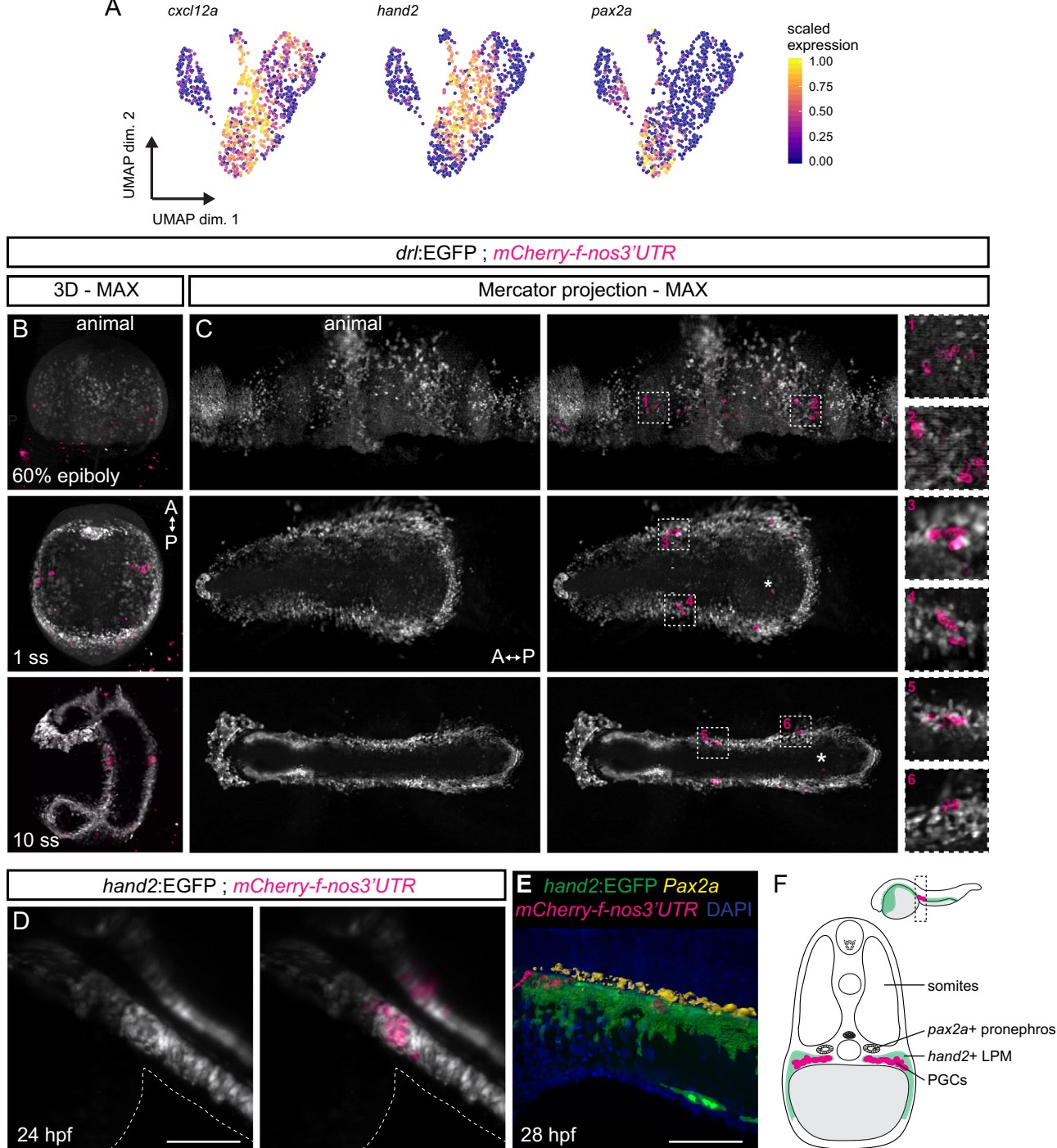

**Fig. 4 Primordial germ cells migrate with, and home within, the *hand2*-positive LPM. A** UMAP plots of *cxcl12a*, *hand2*, and *pax2a*. Cells are colored by scaled expression values using lower and upper 1%-quantiles as boundaries. **B**, **C** Snapshots of time-lapse movie (Supplementary Movie 1) showing *drl:EGFP* with primordial germ cell (PGC) marker *mCherry-f'-nos3'UTR*, with 3D renderings at 60% epiboly, 1 ss, and 10 ss (**B**) and maximum intensity Mercator projections for the same time points (**C**). The PGCs migrate from four different clusters during gastrulation into two bilateral clusters during somitogenesis (boxed regions). Note the sparse (lost) PGCs located within the endoderm (asterisks). Animal side is for 60% epiboly to the top, anterior for 1 ss and 10 ss (**B**) to the top and (**C**) to the left. **D** Imaging of *hand2:EGFP* with *mCherry-f'-nos3'UTR* showing a dorsal view at 24 hpf: the PGC clusters fall completely within the *hand2:EGFP* domain. The outlines of the yolk extension are highlighted. **E** Segmented 3D rendering of a *hand2:EGFP; mCherry-f'-nos3'UTR* double-positive embryo stained with anti-Pax2a (yellow) and DAPI (blue) at 28 hpf. **F** A schematic transverse section of a 24 hpf embryo showing two bilateral PGC clusters (magenta) lateral of the developing gut, ventral-lateral of the Pax2a-positive developing pronephros (yellow) within the *cxcl12a*-high *hand2*-positive LPM/coelomic epithelium (green). Scale bar **D**, **E** 250 μm.

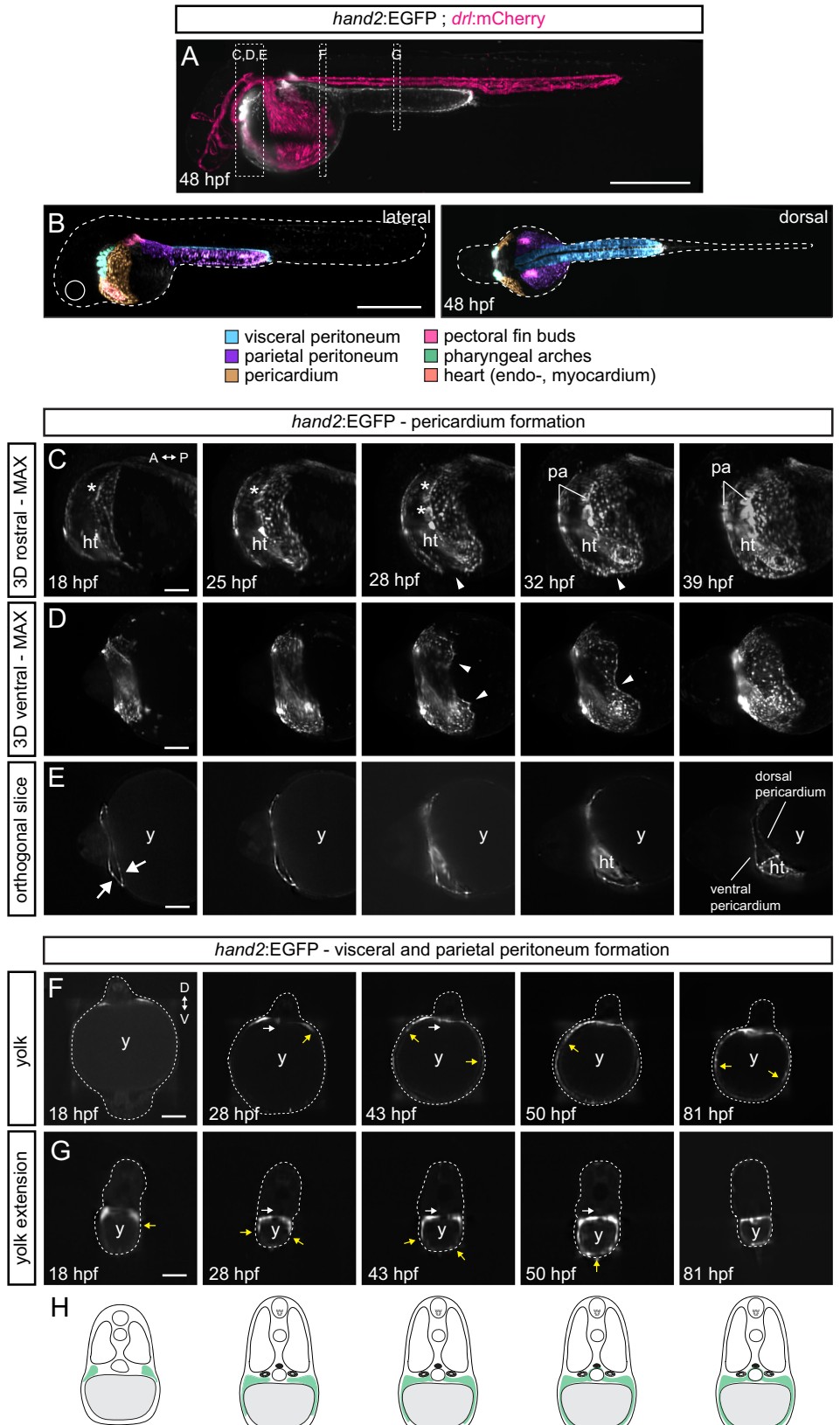

**hand2:EGFP ; *drl*:mCherry**

48 hpf

lateral | dorsal

48 hpf

🟦 visceral peritoneum
🟪 parietal peritoneum
🟧 pericardium
🟥 pectoral fin buds
🟩 pharyngeal arches
🟧 heart (endo-, myocardium)

**hand2:EGFP - pericardium formation**

3D rostral - MAX
18 hpf | 25 hpf | 28 hpf | 32 hpf | 39 hpf

3D ventral - MAX

orthogonal slice

**hand2:EGFP - visceral and parietal peritoneum formation**

yolk
18 hpf | 28 hpf | 43 hpf | 50 hpf | 81 hpf

yolk extension
18 hpf | 28 hpf | 43 hpf | 50 hpf | 81 hpf

number. These data are in line with previous observations that documented i) coelomic epithelial progenitors migrated to the midline in *hand2* mutants but failed to properly wrap around the gut[55], and ii) that *hand2* mutants feature perturbed heart field migration and reduced amounts of cardiac progenitor cells[54,76,118,127]. Of note, we only observed few LPM lineage-labeled cells and *hand2*:EGFP-expressing cells lining the body wall, respectively, indicating that also the ventral migration of the parietal peritoneum progenitors is perturbed in *hand2* mutants (Fig. 6C, F, G–J and Supplementary Fig. 8 and Supplementary Movie 4). Consequently, *han*^S6-homozygous embryos also fail to form a proper body wall, likely resulting in herniation of the yolk

**Fig. 5 Formation of the *hand2*-positive mesothelium.** Multiday, multi-angle SPIM of *hand2:EGFP* zebrafish embryos from 18 to 82 hpf ($n = 6$). **A** Maximum intensity projection (MAX) of embryo expressing the LPM-marking *drl:mCherry* (24 hpf onwards restricted to cardiovascular lineages) (magenta) and *hand2:EGFP* (greyscale) at 48 hpf. Boxes depict selected regions in **C**–**G**. **B** *hand2:EGFP*-positive populations within 48 hpf embryo, lateral view left and dorsal view right. **C**–**E** 3D-rendered *hand2:EGFP* embryo, focused on pericardium formation. **C** Rostral view, visualizing formation of pharyngeal arches (asterisks) and primary heart tube. **D** Ventral view, illustrating left and right flanks of the forming pericardium meeting at the midline (32 hpf, arrowheads). **E** Single plane, highlighting how the pericardial cavity forms within anterior LPM (arrows). *hand2:EGFP* labels coelomic epithelium, contributing to ventral and dorsal pericardium. **F**, **G** Single-plane cross-sections showing the migration of *hand2*-positive cells over yolk (**G**) and yolk extension (**H**), forming parietal peritoneum. Yellow arrows point out dorsal-ventral directed migration path. White arrows indicate inwards-migrating EGFP-expressing cells, contributing to coelomic epithelium maturing into visceral peritoneum. **H** Schematics of how *hand2*-expressing cells laterally migrate over the yolk forming parietal peritoneum, and migrating medially to wrap around endodermal-derived organs forming the visceral peritoneum. Red blood cells (rbc), heart tube (ht), pharyngeal arches (pa), and yolk (y). Scale bars **A**, **B** 500 μm and **C**–**G** 25 μm.

along the anterior–posterior axis over time. These observations in zebrafish connect *hand2* expression and function in dedicated LPM progenitor cells to the proper execution of mesothelial membrane formation.

**Mesothelioma activates early LPM transcription factors.** Upon exposure to asbestos fibers, mesothelial cells within the visceral and parietal pleura can transform into mesothelioma tumors[36]. As the cell(s) of origin have been hypothesized to reside within the mesothelium, we sought to determine whether mesothelioma tumors feature a transcriptional signature akin to the mesothelial progenitors we found in zebrafish (Fig. 2).

First, we revisited our transcriptomics data set obtained from mouse mesothelioma triggered by repeated exposure to crocidolite fibers (blue asbestos)[128] (Fig. 7A). As previously established, genes found upregulated in human mesothelioma development (including *Msln* and *Wt1*) or downregulated (including tumor suppressors *Nf2* and *Bap1*) behaved analogously in this mouse model, both in pre-neoplastic lesions and in fully formed tumors when compared to healthy adult mesothelium (Fig. 7B)[41,42,128–131]. Early LPM lineage markers, including *Scl/Tal1*, *Etv2*, and *Pax2*, were not detected with RNA-seq in the adult mouse mesothelium samples. Notably, the sham-treated mesothelial tissue also expressed no or low levels of transcripts for LPM-associated genes including *Hand2*, *Gata4/5/6*, *Meis3*, *Foxf1*, *Mixl1*, and *Lmo2* (Fig. 7B). In contrast, the transition to pre-neoplastic lesions and to mesothelioma after crocidolite exposure was accompanied by the upregulation of mouse orthologs of several genes associated with mesothelium progenitors in zebrafish: expression of *Hand2*, *Wt1, Gata4/5/6*, and *Meis3* increased in crocidolite-exposed mesothelium, with a particularly striking upregulation in fully formed tumors (Fig. 7B). Also *Foxf1* was upregulated in crocidolite-induced tumors, a gene which has been associated with the developing splanchnic mesoderm in lamprey, chicken, and mouse, but not in zebrafish[13,14,132,133] (Fig. 7B). Following immunohistochemistry for Hand2, we detected specific and prominent nuclear Hand2 immunoreactivity in crocidolite-induced mesothelioma tumor sections as well as a mosaic fraction of Hand2-expressing cells surrounded by Hand2-negative cells in crocidolite-exposed mouse mesothelium (Fig. 7C and Supplementary Fig. 9). Altogether, these observations suggest that the transformative events leading to mesothelioma in a crocidolite-based mouse model are accompanied by the upregulation (or re-initiation) of early LPM genes we found associated with mesothelial progenitors in development (Fig. 2).

We next sought to corroborate whether human mesothelioma also feature *HAND2* expression. Analyzing expression levels of *HAND2* using qPCR in human pleural mesothelioma (MPM) ($n = 36$) compared to healthy pleura ($n = 4$), we observed that individual tumors showed heterogeneous yet consistent upregulation of *HAND2*, with individual tumors expressing high levels of

*HAND2* mRNA ($p = 0.0596$) (Fig. 7D). We further detected nuclear HAND2 immunoreactivity in an epithelioid mesothelioma sample with high *HAND2* mRNA levels (Fig. 7E). To further establish if expression of HAND2 and other mesothelial progenitor-associated LPM genes are re-activated in mesothelioma, we mined the mesothelioma-associated transcriptome data deposited in The Cancer Genome Atlas (TCGA) as generated by the TCGA Research Network ([https://www.cancer.gov/tcga]). We compared gene expression for *HAND2* and other mesothelial progenitor genes with a collection of mesothelioma-associated genes, and several tissue-specific and ubiquitous house-keeping genes. We found that increased or re-activated *HAND2* expression was not an obvious signature across the analyzed MPM samples, but rather a feature of a subset of tumors across the epithelioid, sarcomatoid, and biphasic classifications (Fig. 7F). *HAND2* expression coincided with differential expression of the previously mesothelium- and mesothelioma-associated genes *MSLN* and *WT1*, as well as with *GATA5* and *MEIS2* (Fig. 7F). Notably, while we found in zebrafish mesothelial progenitors *meis3* as the most-prominent *MEIS* family gene co-expressed with *hand2* (Fig. 2), its redundant paralogs *meis2a* and *meis2b* were also enriched in *hand2*-expressing zebrafish progenitors (Supplementary Fig. 4)[134]. This *HAND2*-expressing group of mesothelioma tumors did not cluster notably with tumors featuring differential expression of the common mesothelioma-associated tumor suppressor genes *BAP1* and *CDKN2A* (Fig. 7F and Supplementary Fig. 9 and Supplementary Table 1). Taken together, these observations indicate that mesothelioma tumors can upregulate or reactivate several developmental factors associated with a mesothelium progenitor state, in particular Hand2 that in zebrafish provides the earliest mesothelium progenitor marker. We postulate that, following malignant transformation, mesothelioma tumors acquire features of a developmental progenitor program deployed by the early LPM to specify mesothelium progenitors, endowing transformed cells with unique migratory and lineage properties.

## Discussion
Despite their various contributions to development and homeostasis, the early embryonic origins of mesothelia and their genetic control have remained uncertain. Uniquely positioned within the developing embryo to cover inner organs and the forming body cavities, our results uncover that mesothelial progenitor cells emerge among the earliest distinguishable LPM cell fates following *hand2* expression.

Our finding that expression of the transcription factor Hand2 demarcates the merging mesothelial progenitors within the post-gastrulation LPM provides means to study the mesothelium in development and disease. *Hand* gene function centrally contributes to cardiac, limb, and pharyngeal arch development across several models[51,54,57,135,136]. Our results indicate that *Hand* genes also act as conserved regulators of mesothelial fates.

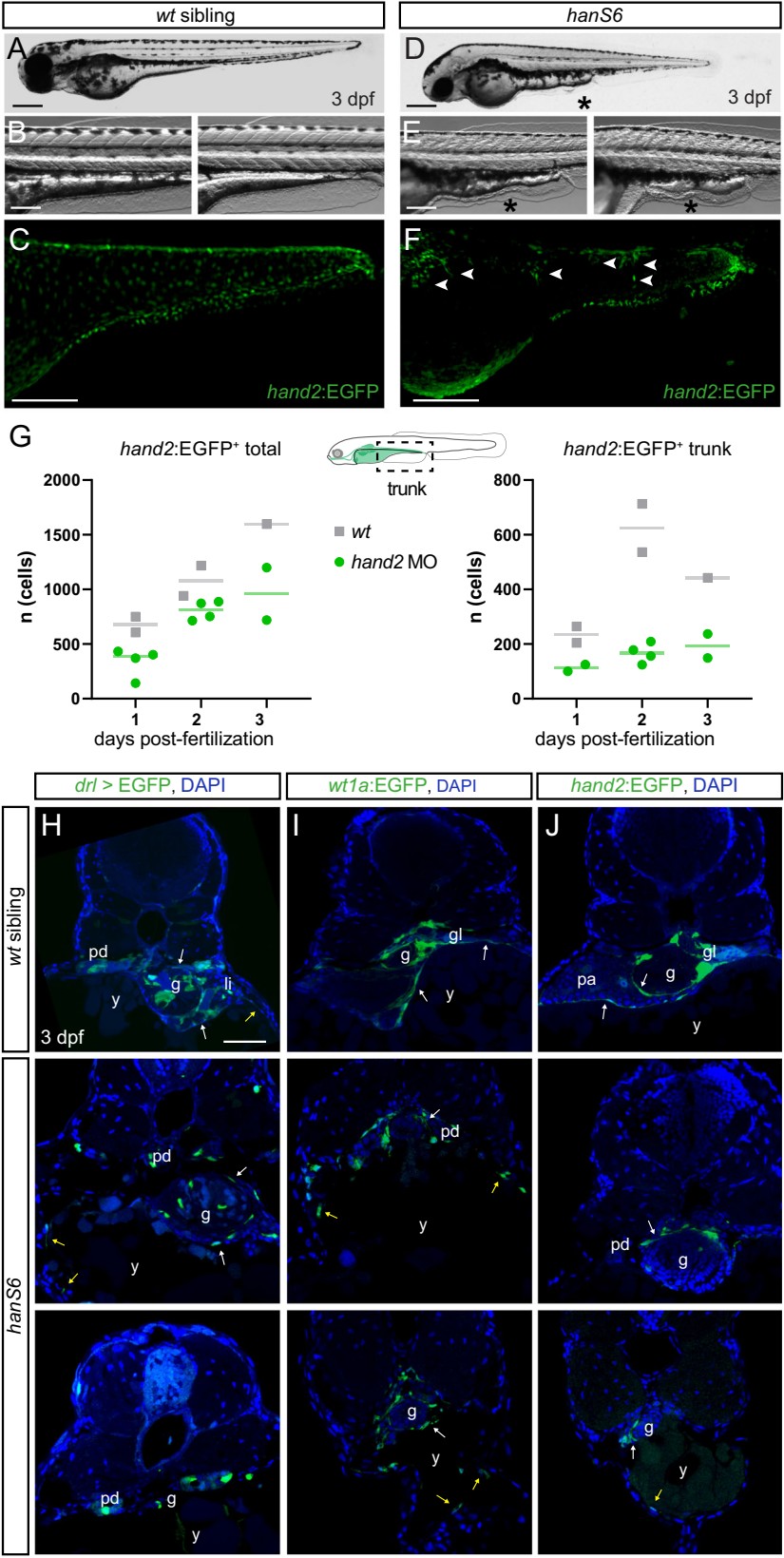

Gene expression studies have defined *hand2* as a marker for the forming body wall in tilapia and medaka, and have found *Hand1/2* in the flank of the developing mouse embryo[53,122,137,138]. Complementing studies in mice have demonstrated that *Hand1*-expressing progenitors contribute to cell layers of the gut including smooth muscles[53,96,122]. A possible ancient association of *Hand* genes with the LPM and thus mesothelium progenitors followed the expression of amphioxus *AmpiHand* and lamprey *LjHandA* within their presumptive LPM[13]. While zebrafish have seemingly lost their *Hand1* ortholog[139], our data suggest an early role for Hand2 and Hand factors in general in the differentiation of mesothelial progenitors from the LPM (Fig. 3).

**Fig. 6 Loss of *hand2* causes peritoneum defects. A–E** Lateral views of phenotypically wildtype **A–C** zebrafish embryos and representative *han^S6* mutants (**D–F**) at 3 dpf. Note the abnormal phenotype of the yolk extension (representative reference wildtypes in **B**) in *han^S6* mutant embryos (asterisks in **D**, **E**); the increasing autofluorescence of the ventral yolk is possibly caused by the progressive disorganization of the yolk tissue. SPIM-based live imaging of *hand2:EGFP* transgene expression that depicts mesothelial progenitors over the posterior yolk and yolk extension (wildtype reference in **C**) documents disorganized and only partially migrated mesothelial progenitors (arrowheads in **F**). **G** Quantification of *hand2*:EGFP-expressing cells in wildtype versus *hand2* MO-injected embryos (morphants) over time (days post-fertilization). The left graph depicts the total amount of *hand2*:EGFP+ cells in an embryo (including heart and pectoral fin field), the right graph the *hand2*:EGFP+ in the trunk region (boxed area in schematic). See Supplementary Fig. 8 for Imaris analysis. Source data are provided as a Source Data file. **H–J** *han^S6* in the background of *drl:creERT2;hsp70l:Switch* (**H**), *wt1a:EGFP* (**I**), and *hand2:EGFP* (**J**). Visceral peritoneum is disorganized around the internal organs (pointed out with white arrows) and the body wall and the associated parietal peritoneum are malformed (pointed out with yellow arrows) in *han^S6* mutants compared to wildtype controls. Gut (g), liver (li), pancreas (pa), pronephric duct (pd), and yolk (y). Nuclei in blue (DAPI). Scale bars **A**, **D** 250 μm, **B**, **C**, **E**, **F** 125 μm, and **H–J** 50 μm.

Several fates emerging from the LPM stem from multi-lineage progenitors, such as the *Scl/Tal1*-positive hemangioblasts that contribute to endothelial and hematopoietic lineages[140–143]. Our single-cell transcriptomics analysis of the early *drl* reporter-expressing LPM now captured a surprising heterogeneity within the forming LPM already at the end of zebrafish gastrulation, including putative cardiopharyngeal, endothelial/hematopoietic, and here defined mesothelial progenitor clusters (Fig. 2). These data complement and expand prior scRNA-seq in mouse, zebrafish, and tunicate embryos that captured numerous mesendodermal and mesodermal cell fate territories at various developmental stages[67–71,78]. With *wt1a/b* expression appearing later than *hand2* and seemingly restricted to the visceral mesothelial layers (Fig. 3), *hand2* provides the earliest and most complete marker for mesothelial progenitors in zebrafish and possibly beyond. Our findings are applicable to in vitro differentiation protocols for multi-lineage organoids with desirable stromal components, such as to guide pluripotent stem cell differentiation into representative cell types of the gastro-intestinal tract. Further, our work provides a framework to interpret cell types deemed mesenchymal in numerous systems.

Our results provide several insights into mesothelial progenitor biology. Combined expression of *hand2* with *sfrp5*, *meis3*, and *foxh1* potentially defines a rudimentary mesothelial progenitor signature in the early LPM (Fig. 2E–K and Supplementary Fig. 4); nonetheless, our data could also possibly indicate progressively more differentiated mesothelial progenitors within our scRNA-seq-captured populations (within the timeframe staged of embryo collection). Individually, however, these genes have all previously been linked to mesothelial biology. *sfrp5* expression has been reported in mesenchymal cells around internal organs in zebrafish and in the foregut-surrounding epithelium in *Xenopus*[144,145]. *meis3* expression has been linked to mesenchymal cells lining the developing intestine and to enteric neuron migration in zebrafish[123,146]; we note that the potentially redundant *meis2a/b* are also expressed in the early LPM and mesothelial progenitors (Supplementary Fig. 4). During segmentation stages, *foxh1* is involved in controlling left-right asymmetry in response to Nodal signaling in a lateral LPM domain of so-far unassigned fate[89,147]; our data now proposes that this activity of *foxh1*, and possibly the response to left-right cues, is confined to the *hand2*-positive mesothelial progenitors (Fig. 2H).

In addition, we observed that the PGCs migrate already from gastrulation stages associated with the mesothelium-primed LPM that establishes their ventral migration border (Fig. 4). Together with the kidney that borders their migration dorsally, our documentation of *hand2*-expressing mesothelium as ventral migration boundary provides additional context to the migration of PGCs to their final destination where the future gonads will form. Recent scRNA-seq studies of early mouse embryos have uncovered *Hand2*-expressing cell clusters that were deemed mesenchymal, and that we now likely recognize as mesothelial progenitors

by their complement of LPM-associated transcripts (Supplementary Fig. 5)[28,99]. Altogether, these observations connect the coelomic epithelium progenitors to key aspects of body plan organization already in early development. The exact composition of a mesothelial progenitor-controlling program and its evolutionary conservation beyond the genes uncovered here warrants further attention.

Expanding previous loss-of-function analyses that documented *Hand* gene contribution to heart and forelimb development[51,54,148,149], we here add that mesothelial progenitors fail to properly migrate and to establish normal cell number in *hand2*-mutant zebrafish, resulting in ventral herniation of the yolk at 3 dpf due to incomplete closure of the ventral body wall (Fig. 6 and Supplementary Fig. 8 and Supplementary Movie 4). How loss of *hand2* leads to dysregulated migration and reduced cell number of the mesothelial progenitors warrants future efforts. In congenital malformations affecting the body wall including omphalocele and gastroschisis, the future ventral abdominal body wall fails to form properly, resulting in internal organ protrusion and herniation of abdominal muscles, among other phenotypes[15,150,151]. Among the candidate factors involved in this process is TGF-β signaling that has been repeatedly associated with Hand2 function and midline migration[55,58,121], as well as Six4 and Six5[152]. Further, Hand2 regulates extracellular matrix components including repression of Fibronectin production, disruption of which lead to migration defects in the heart and prospective visceral mesothelium[55,153]. Hand2 loss however also clearly impacts the number of cardiac progenitor cells[54,118], yet the underlying mechanism remains uncertain. While not exhaustive, our characterization of mesothelial phenotypes in *hand2* indicates that migration and possible proliferation and cell survival defects upon *hand2* loss also occur in the developing mesothelium. Altogether, our data indicate that *hand2* is not only active early on during mesothelium emergence but also functionally contributes to correct mesothelium formation in zebrafish.

Asbestos-induced mesothelioma remains a global health challenge[36,39]. While asbestos fibers have been causally linked to pleural mesothelioma[37], the transformative mechanism and the exact cell of origin await clarification. Different patient stratifications have been assigned to mesothelioma with biphasic, epithelioid, or sarcomatoid tumors based on histopathology[36]. Nonetheless, only a few genetic markers are known, including YAP activity and altered BAP1 expression, and their role in mesothelioma formation and maintenance remains unclear[36,42]. Work in mouse models has uncovered activation of stemness-related signaling pathways and an increase in *Msln*-expressing cells upon asbestos exposure[128]. Human epithelioid mesothelioma show upregulated expression of coelomic epithelium-associated genes, including *WT1*[154,155]. Notably, the non-coding RNA *Fendrr* (*Fetal-lethal noncoding*) is strongly upregulated in mouse and human epithelioid mesothelioma and shares a

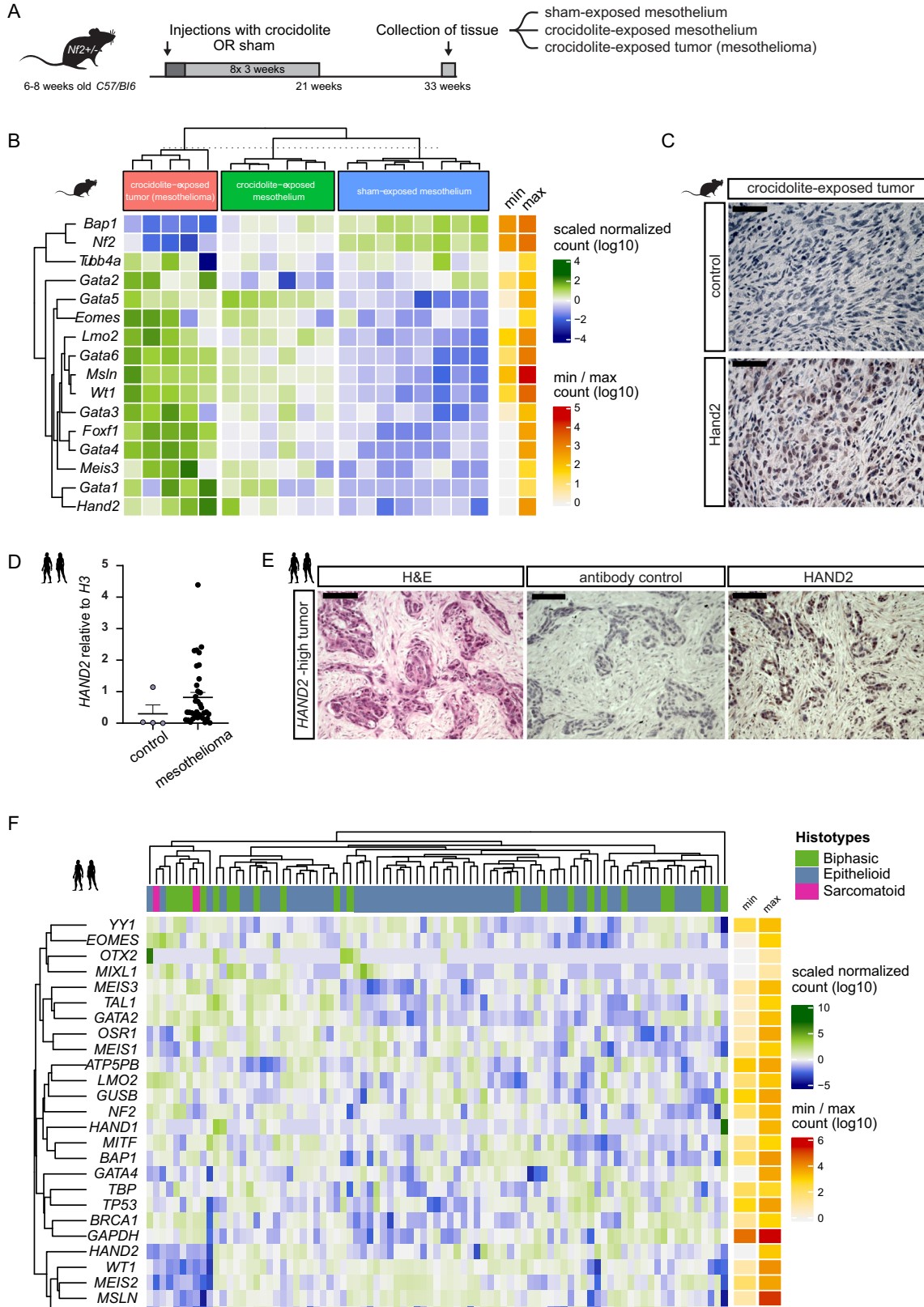

bidirectional promoter with the LPM transcription factor gene *Foxf1*[41,156,157]. These observations indicate that at least some populations of mesothelioma cells feature a gene expression profile reminiscent of late developmental markers for the coelomic epithelium.

Aberrant reactivation of early developmental programs is increasingly recognized as contributor to tumorigenesis[158–160]. Extrapolating from our developmental findings in zebrafish, our data propose that mesothelioma might reactivate LPM-specific factors involved in initial mesothelial progenitor formation: in

**Fig. 7 Mesothelioma reactivates an early LPM program. A** Modeling loss of mesothelial homeostasis during mesothelioma tumorigenesis: 6–8 weeks-old *Nf2+/−* C57Bl/6J mice exposed to crocidolite intra-peritoneally (400 μg) every 3 weeks with eight total treatments, parallel to controls. 33 weeks after initial exposure, tissue was collected. **B** Heatmap of LPM-associated genes, either upregulated (*Msln, Wt1*) or downregulated (*Nf2, Bap1*) mesothelioma genes, and negative control (*Tubb4a*). Bins colored by row-scaled log2-normalized counts; columns (samples) split by treatment group; rows and columns ordered by hierarchical clustering (scaled expression values). Right side row indicates raw count range. **C** Immunoreactivity of Hand2 protein in mouse mesothelioma. **D** Relative expression of *HAND2* by qPCR in normal pleura (*n* = 4) and human malignant mesothelioma (*n* = 36); trend (*p* = 0.0596) towards increased *HAND2* levels in tumors (Mann–Whitney two-tailed test, exact *p*-value). Mean ± SE. **E** Immunoreactivity of HAND2 protein in *HAND2*-high human mesothelioma. **F** Heatmap representing unsupervised hierarchical clustering of human mesothelioma from TCGA RNA-seq (*n* = 87), depicting expression of mesothelial progenitor-associated LPM genes, mesothelioma-associated genes (*WT1, MSLN, BAP1, NF2,* and *TP53*), unrelated control (*YY1, OTX2,* and *BRCA1*) and housekeeping genes (*ATP5PB, GUSB, TBP,* and *GAPDH*). Columns represent histotypes, rows represent log10-transformed, batch-normalized mRNA expression levels of mesothelioma samples. Scale bars **C** 50 μm, and **D** 100 μm.

both mouse and human mesothelioma, we found the expression of Hand2 and additional transcription factor genes associated with early mesothelium-primed LPM (Figs. 2 and 7). Notably, crocidolite-exposed tissue showed intermediate expression levels for mesothelial progenitor genes, further supporting a model wherein mesothelioma formation involves a temporal re-activation of LPM-expressed genes as deployed in initial meso-thelium development (Fig. 7). While more work is needed to establish whether and which LPM transcription factors contribute to mesothelioma formation and maintenance, our findings provide several mRNA transcript and protein expression assays to potentially further stratify mesothelioma in the clinic. We speculate that the malignant transformation of adult mesothelium, and in particular Hand2, endows altered cells with features that recapitulate the migration, proliferation, and stemness that distinguished mesothelial progenitor cells already during their initial development in the LPM.

## Methods

All research described herein complies with all relevant ethical regulations as reviewed and approved by the University of Colorado School of Medicine and the University of Zurich.

**Zebrafish husbandry.** Zebrafish (*Danio rerio*) husbandry and experiments were performed according to the European Communities Council Directive (86/609/EEC), the recommendations of the Swiss authorities (Animal Protection Ordinance), and according to IACUC regulations at the University of Colorado School of Medicine, Anschutz Medical Campus. Protocols and experiments were performed as approved by the cantonal veterinary office of the Canton Zurich (Kantonales Veterinäramt, permit no. 150, TV4209), in accordance with the European Union (EU) directive 2011/63/EU as well as the German Animal Welfare Act, and as by the IACUC at the University of Colorado School of Medicine, Anschutz Medical Campus (protocol no. 00370 and no. 00979). All zebrafish were raised, kept, and handled according to established protocols[161] if not noted otherwise.

**Transgenic zebrafish lines and transgene activity.** Established transgenic and mutant lines used in this study includes *Tg(drl:EGFP)cz333[159]*, *Tg(drl:mCherry)zh705[162]*, *TgBAC(hand2:EGFP)pd24[163]*, *Tg(pax2.1:EGFP)e1[164]*, *Tg(lmo2:loxp-DsRed-loxp-EGFP)rj2[85]*, *Tg(-6.8wt1a:EGFP)[105]*, *Tg(wt1b:EGFP)li1[105]*, *Tg(kop:mCherry-f′-nos3′UTR)er1[112]*, *Tg(drl:creERT2;alpha-crystallin:YFP)cz333[59]*, *Tg(-6.8wt1a:creERT2)cn10[106]*, *Tg(−3.5ubb:loxP-EGFP-loxP-mCherry)cz1701[165]*, *Tg(-1.5hsp70:loxP-STOP-loxP-EGFP;alpha-crystallin:Venus)zh70[161]*, and *hand2* mutants (*hanS6*)[54].

The construct to generate the transgenic line *TgBAC(wt1b:rtTA-p2A-creERT2)cn19* (referred to in the text as *wt1b:creERT*) was generated by BAC recombineering using EL250 bacteria[166]. Fragments were amplified by PCR, adding 50 nucleotide homology arms. First, the *iTol2Amp-γ-crystallin:RFP* cassette[162] was amplified using primers 1. *pTarBAC_HA1_iTol2_F 5′-gcgtaagcggggcacatttcattacctctttctccgcacccgacatagatCCCTGCTCGAGCCGGGCCCAAGTG-3′* and *pTarBAC_HA2_iTol2CrystRFP_R 5′-gcg gggcatgactattggcgcgccggatcgatccttaattaagtctactaTCGAGGTCGACGGTATCGATTAAA-3′* and recombined into the backbone of the BAC clone CH73-186G17 to replace the BAC-derived *loxP* site. Then, the *rtTA-p2A-iCreERT2* cassette was amplified and recombined replacing the *ATG* of the *wt1b* coding sequence, with primers *wt1b_HA1_rtTA_F 5′-gacattttgaactcagatattctagtgttttgcaacccagaaaatccgtcACCATGGT CGACGCCACAACCAT-3′* and *wt1b_HA2_FRT_R 5′-gcgctcaggtctctgacatccgatcca tcgggccgcacggctctgtcagGGAGGCTACCATGGAGAAG-3′*. Finally, the Kanamycin resistance cassette was removed by inducing expression of Flipase recombinase in the

EL250 bacteria. The final *BAC* was purified with the HiPure Midiprep kit (Invitrogen) and injected along with synthetic *Tol2* mRNA into wildtype strain zebrafish embryos. Sequence information and primer details are freely available upon request.

Six to eighteen somite staged (ss) dual-fluorescent embryos were imaged using a Leica SP8 upright confocal microscope with HCX-Apochromat W U-V-I 20×/0.5 water correction objective.

**Mouse transgenic strains.** The mouse strains used in this study are *Han-d1EGFPCreΔNeo/+[96]*, *Hand2Cre (dHand-Cre)[97]*, and the *ROSA26 Reporter (R26R)[98]*. The *Hand2Cre* transgenic strain was produced using a 7.4 kb genomic fragment from the mouse *Hand2* locus immediately upstream of the transcriptional start site. The *Hand1EGFPCreΔNeo/+* strain was produced by knocking a *Cre* cassette into the mouse *Hand1* locus. Abdominal transverse sections of E14.5 and E15.5 mouse embryos lineage-labeled by β-galactosidase (LacZ) expression off *R26R* was performed essentially as per previous analyses of *Hand1/2*-based lineage tracing[51]: briefly, embryos were dissected from the uterus and fixed in 2% paraformaldehyde (PFA) with 0.2% glutaraldehyde for up to 1 h, followed by rinsing in PBS and incubation PBS with 5 mM K₃Fe(CN)₆, 5 mM K₄Fe(CN)₆;3H₂0, 2 mM MgCl₂, and 1 mg/ml X-gal overnight. Rinsing embryos with PBS stopped the staining reaction, followed by post-fixation in 4% PFA. Labeled embryos were dehydrated in ethanol and embedded in paraffin for sectioning. Abdominal transverse sections were cut at 10 μm intervals and dried on microscope slides. Paraffin was removed with xylene before sections sections were stained with H&E (or simply treated with cytoseal) before application of the cover slip.

**Human tumor specimens.** Mesothelioma tumor specimens were collected from MPM patients treated at the Department of Medical Oncology and Department of Thoracic Surgery between January 2007 and December 2014. Non-tumor pleural tissue was received from four patients undergoing mesothelioma-unrelated thoracic surgery. The Zurich Cantonal Ethics Committee approved the study under the reference number StV 24-2005 and 29-2009, and informed consent was collected from all patients. Immediately upon receiving the corresponding non-tumoral resections and tumor specimens collected from diagnostic biopsies prior to neoadjuvant chemotherapy, one part of the samples was processed for total RNA extraction and another part was fixed in 1× PBS 4% PFA for paraffin embedding.

**Zebrafish CreERT2-based lineage tracing.** Genetic lineage tracing experiments were performed using the Cre/loxP recombination system with 4-OHT-inducible creERT2 transgenic lines. *hsp70l:Switchzh701* reporter zebrafish[61] were crossed with *creERT2* drivers *drl:creERT2[59]*, *wt1a:creERT2*, and *wt1b:creERT2*. Lineage-labeling was induced at indicated time points using fresh or pre-heated (65 °C for 10 min) 4-Hydroxytamoxifen (4-OHT) (Sigma H7904) stock solutions in DMSO at a final concentration of 10 μM in E3 embryo medium[167]. To activate the GFP transcription in *hsp70l:Switch*, embryos were incubated at 37 °C for 1 h and 2–3 h before fixation. Subsequently, the embryos were fixed in 4% PFA at 4 °C overnight, washed and stored in PBS, and processed for imaging.

**Zebrafish histology and sectioning.** Transverse sections were performed according to previous work[58], in detail: 4% PFA-fixed zebrafish embryos were embedded in 6% low-melting agarose (LMA, Sigma-Aldrich) dissolved in 0.1% Tween-20 (Sigma-Aldrich) in PBS (PBS-T) and dissected into 130 μm slices with a vibratome (Leica VT 1000S). For immunostaining, the sections were incubated for 30 min in permeabilization buffer (1% BSA (AMRESCO), 1% DMSO (Sigma-Aldrich), and 0.3% Triton X-100 (Sigma-Aldrich) in PBS) and washed with PBT (0.1% Triton X-100 and 1% BSA in PBS). Antibodies were dissolved in PBT and incubated overnight at 4 °C (primary antibody) or for 4 h at room temperature (secondary antibody). Primary antibody used was α-SM22 alpha (TAGLN) antibody (AbCam, ab14106, 1:250). Secondary antibody used was goat-anti-rabbit Alexa594 (Alexa Fluor, Life Technologies, A-11012, 1:300). Sections were mounted with DAPI-containing Vectashield (Cat#H-1200, Vector Laboratories). Images were obtained with the Zeiss LSM710 confocal microscope equipped with Plan-Apochromat 40×/1.3 oil DIC M27 or 40×/1.2 immersion correction DIC M27

objectives. Images were adjusted for brightness and cropped using ImageJ/Fiji(1.52q 13, Java 8)[168].

Embryos for whole-mount immunostainings were processed as described above for the sections. Primary antibodies used were anti-Pax2 (rabbit, GeneTex, GTX128127, 1:250) and anti-Myosin heavy chain (mouse, DSHB, MF20, 1:20). Secondary antibodies (1:300-1:400) used were goat-anti-rabbit Alexa633 (Alexa Fluor, Life Technologies, A-21070) and goat-anti-mouse Alexa594 (Alexa Fluor, Life Technologies, A-11032). Embryos were either imaged on a Leica SP8 upright confocal microscope or Zeiss Z.1 light sheet microscope (see details below).

**Gene expression analysis and in situ hybridization (ISH)**. First-strand complementary DNA (cDNA) was generated from pooled zebrafish RNA isolated from different developmental stages using SuperScript III first-strand synthesis kit (Invitrogen). DNA templates were generated using first-strand cDNA as a PCR template and the primers as specified for each gene of interest; for in vitro transcription initiation, the T7 promoter 5′-*TAATACGACTCACTATAGGG*-3′ was added to the 5′ ends of reverse primers. Specific primers per gene were *meis3*: forward 5′-*TACCACAGCCCACTACCCTCAGC*-3′, reverse 5′-*TAATACGACTCACTATAGGGTCAGCAGGATTTGGTGCAGTTG*-3′; *gata5*: forward 5′-*CACCATGTATTCGAGCCTGGCTTTATCTTCC*-3′, reverse 5′-*TAATACGACTCACTATAGGGTCACGCTTGAGACAGAGCACACC*-3′; *sfrp5*: forward 5′-*GAATCACAGCAGAGGATG*-3′, reverse 5′-*TAATACGACTCACTATAGGGCATCTGTACTAATGGTCG*-3′; *cdx4*: 5′-*CACCATGTATCACCAAGGAGCG*-3′, reverse 5′-*TAATACGACTCACTATAGGGTAAATCCACAACCCACGCC*-3′. PCR reactions were performed under standard conditions as per manufacturer's protocol using Phusion high-fidelity DNA polymerase (Thermo Fisher Scientific). RNA probes were generated via overnight incubation at 37 °C using 20 U/μL T7 RNA polymerase (Roche) and digoxigenin (DIG)-labeled dNTPs (Roche) as per manufacturer's protocol. The resulting RNA was precipitated in lithium chloride and EtOH. Wildtype strain zebrafish embryos were fixed in 4% PFA in PBS overnight at 4 °C, dechorionated, transferred into 100% MeOH, and stored at −20 °C. ISH of whole-mount zebrafish embryos was performed essentially as per standard protocols[169].

**RNAscope assay**. The RNAscope assays were carried out based on a previously published protocol[170] and the manufacturer's instruction (Advanced Cell Diagnostics). Wildtype strain or transgenic zebrafish embryos of tailbud stage until early somitogenesis were fixed in 4% PFA in PBS for 1 h at RT or overnight at 4 °C, hand dechorionation, and transferred into 100% MeOH and stored at −20 °C. Protease digestion of embryos using Protease III was performed for 20 min at RT followed by rinsing the embryos three times in 0.01% PBS-T (0.01% Tween-20). Target probe hybridization was performed overnight at 40 °C. Target probes were designed by Advanced Cell Diagnostics for *hand2*, *foxh1*, and *meis3*. For RNA detection, incubation with the different amplifier solutions was performed in an incubator at 40 °C. After each hybridization step, the embryos were washed three times with 0.2× SSCT for 15 min. The embryos were then incubated with ACD's DAPI solution overnight at 4 °C. Prior to imaging, embryos were rinsed in 0.01% PBT.

**Tumor RNA extraction, cDNA synthesis, and qPCR**. RNA extraction, cDNA synthesis, and qPCR were performed as previously described[171]: in detail, upon RNA extraction using Qiagen RNAeasy Minikit (Qiagen, ID: 74106), 400–500 ng of RNA was used to perform reverse transcription using QuantiTect Reverse Transcription Kit according to manufacturer's protocol. To determine the quantitative expression of *HAND2* in MPM and non-tumoral specimens, obtained cDNA was used in SYBR-Green PCR assay and reactions were run on 7500 Fast Real-time PCR system thermal cyclers (Applied Biosystems). The following primers were used: *HAND2* forward 5′-*CAGGACTCAGAGCATCAACAG*-3′ and reverse 5′-*TCCATGAGGTAGGCGATGTA*-3′. Normalization was performed based on the expression of Histone H3 detected with primers 5′-*GGTAAAGCACCCAGGAAGCA*-3′ and 5′-*CCTCCAGTAGAGGGCGCAC*-3′[172]. Specificity of the yielded PCR product was verified by running on a 4% agarose gel and by analyzing corresponding melting curve. The relative gene expression was determined by comparing the PCR threshold between *HAND2* and histones (Ct*HAND2* - Ct*H3* = ΔCt) followed by comparing the ΔCt between tumors and non-tumor samples (ΔΔCt)[172]. Data were analyzed using Mann–Whitney test (two-tailed test).

**Tumor immunohistochemistry**. Tumor tissue sections were de-paraffinized and antigen retrieval was achieved with sodium citrate (0.01 M, pH 6.0) for antigen retrieval, followed by quenching in 0.3% $H_2O_2$ for 20 min and permeabilization in 0.05% Saponin (Fluka) for 5 min. The sections were blocked with 2% BSA/1% horse serum in 1× PBS at RT and incubated with HAND2 antibody (A-12, Sc-398167, dilution 1:100) overnight at 4 °C. Samples were washed with 1× PBS and incubated with the secondary universal antibody (Vectastain, Vector laboratories Inc., Burlingame, CA, USA, PK-6200; prepared as per manufacturer's manual and kit reagents using 100 μL blocking serum stock in 5 mL buffer with additional 100 μL of secondary antibody stock as working solution), for 45 min at RT. To visualize the binding of the HAND2-antibody, 3,3′-diaminobenzidine tetrahydrochloride (DAB, Sigma-Aldrich) was used followed by counterstained with Vector ® Hematoxilyn QS (Vector Laboratories). Images were captured using a

Leica DM6000B light microscope. Control staining was performed by omission of the primary antibody. Tumor tissue sections were de-paraffinized and antigen retrieval was achieved with sodium citrate (0.01 M, pH6) for antigen retrieval, followed by quenching in 0.3% $H_2O_2$ for 20 min and permeabilization in 0.05% Saponin (Fluka) for 5 min. The sections were blocked by incubating with 2% BSA 1× PBS supplemented with 1% horse serum at room temperature and incubated with HAND2 antibody (Santa Cruz Biotechnology A-12, Sc-398167, at dilution 1:100) overnight at 4 °C. Samples were than washed with 1× PBS and incubated with the secondary universal antibody (Vectastain, Vector laboratories Inc., Burlingame, CA, USA, PK-6200), for 45 min at room temperature, according to manufacturer instructions. To visualize the binding of the anti-HAND2 antibody, 3,3′-diaminobenzidine tetrahydrochloride (DAB, Sigma-Aldrich) was used followed by counterstained with Vector Hematoxilyn QS (Vector Laboratories)[173]. Images were captured with a light microscope (Leica DM6000B). Control staining was performed by omission of the primary antibody[173].

**Light sheet sample preparation, microscopy, and image analysis**. Embryos used for long-term imaging or imaging after 1 dpf were treated with 0.003% 1-phenyl-2-thiourea (PTU, Sigma-Aldrich) to prevent melanin pigment formation.

*Panoramic SPIM for* hand2:EGFP;drl:mCherry *imaging during somitogenesis.* At 50% epiboly, embryos in the chorion were mounted in an FEP tube into 1% LMA. Up to six embryos were positioned on top of each other with a minimum gap between them. The FEP tube was mounted in the microscope imaging chamber filled with E3 medium. Time-lapse acquisition was performed in a 4-lens SPIM set up by a standardized image acquisition pipeline[174]. The microscope was equipped with four identical Olympus UMPlanFLN 10×/0.3 NA detection objectives. The subsequent real-time image processing, registration of time points, and 2D map (Mercator) projections were performed with previously described and publicly available Fiji scripts[174]. Briefly, a Z-stack was acquired from every embryo with an interval of 2 min for a period of 14–17 h. Real-time processing of the raw image date was performed during the time-lapse experiment creating radial projections of the Z-stacks, which significantly reduced the amount of raw data amount and also allowed fast image acquisition. The radial projections were post-imaging used for fusion, registration, and map projections. Map projections serve the purpose to visualize the entire surface of a spherical object and to preserve information about angles, distances, areas, and shapes. We chose for the Mercator projection, allowing cell migration analysis due to its angle-preserving properties. Due to the projections, the scale bars are not applicable anymore. Images were processed using ImageJ/Fiji (1.52q 13, Java 8).

*Multidirectional SPIM (mSPIM) for long-term* hand2:EGFP;drl:mCherry *imaging.* To prevent movement of zebrafish embryos during imaging, we injected 30 pg of α-bungarotoxin mRNA[175] at the 1-cell stage. At the 20 ss, embryos that showed no sign of muscular contractions were selected and mounted in FEP tubes filled with 0.1% LMA. A detailed mounting protocol can be found in Daetwyler et al.[116]. Long-term time-lapse imaging was performed using a custom multidirectional SPIM (mSPIM) setup with multi-sample capacity[116]. The microscope was equipped with two Zeiss 10×/0.2 NA illumination objectives and an Olympus UMPlanFL N 10×/0.3 NA detection objective. Up to four embryos were imaged every 20 min for up to 4 days starting around 20 ss from four different angles. Custom-made data processing tools were used for automated data processing, including stitching and fusion of the fluorescent images[116]. Briefly, the acquired images were projected using maximum intensity projections and were stitched and fused with linear blending, and initialized using the translational stage positions of the mSPIM. Subsequently, the stitching parameters of the projections were applied to the 3D data to generate one 3D fused stack per timepoint, angle, channel, and fish. For successful fusion and deconvolution of the four individual angles per embryo, we manually registered them, and then applied the Fiji *Multiview Reconstruction* and *BigStitcher* plugins[176–178] for precise registration and deconvolution. ImageJ/Fiji (1.52q 13, Java 8) and Imaris (9.5, 9.6, and 9.7, Bitplane AG, Zurich, Switzerland) were used for visualization.

*Zeiss Z.1 for all other light sheet imaging.* The Zeiss Z.1 microscope equipped with a Zeiss W Plan-Apochromat 20×/0.5 NA objective was used for all other light sheet microscopy and accordingly mentioned in the figure legends. Embryos were either within or out of the chorion embedded in 1% LMA in respectively a 50 or 20 μL glass capillary. Alive embryos older than 16 ss were additionally mounted with 0.016% ethyl 3-aminobenzoate methanesulfonate salt (Tricaine, Sigma) in the LMA and added to the E3 medium to prevent movement during imaging. For live imaging of the heart, the heartbeat was stopped with 30 mM 2,3-butanedione monoxime (BDM, Sigma). For the multi-angle imaging data sets: we manually registered the four individual angles per embryo and then applied the Fiji *Multiview Reconstruction* and *BigStitcher* plugins[176–178] for fusion and deconvolution of the images[176–178]. Images and movies were further processed using ImageJ/Fiji[168] (as described above) and Imaris (9.5, 9.6, and 9.7). The movies were annotated using standard ImageJ/Fiji tools and plugin *SPIM_DrawArrowInMovie*[179].

Cell quantifications were performed on *hand2*:EGFP transgenic embryos injected with the previously validated[49,123] *hand2* antisense morpholino (*MO2-hand2* 5′-*CCTCCAACTAAACTCATGGCGACAG*-3′) at the 1-cell stage at 1:2

dilution (1 nL). The embryos were raised at 28 °C and imaged in the Z.1 as above at 1, 2, and 3 dpf, alongside uninjected *hand2:EGFP* controls. In Imaris, the baseline and background was subtracted and spots rendering was performed to automate cell picking and counting. Data was plotted using GraphPad Prism (9.3.0.).

*Map projections for* drl:EGFP;kop:mCherry-F-actin-nanos-3′UTR. All pre-processing steps and map projection were performed using custom-developed algorithms in MATLAB (9.7). The overall size and shape of the fused embryo were first determined in an image that was binned four times to increase the processing speed. The binned image was binarized using an adaptive threshold (imbinarize function with a sensitivity of 0.4), and the isosurface of the resulting image was determined. A sphere fit was then performed using linear least squares with the ellipsoid_fit function from Mathworks ([https://ch.mathworks.com/matlabcentral/fileexchange/24693-ellipsoid-fit]). The center and radius of the sphere were then used for performing Mercator projection of the fused images. The rational for using Mercator projections in described above under *Panoramic SPIM*.

For each fused image, a multi-layered projection was performed by generating concentric circles with different radii (step size: 2 μm). This allowed for unwrapping different layers of the 3D fused embryo onto 2D surfaces. For each layer of the map projection, a dummy image in polar coordinates was first generated that extended from −90° to +90° in latitude and 0° to 360° in longitude. Using Mercator projection formulas, the latitudes and longitudes that correspond to each position in the projected map were then determined. The cartesian (*x,y,z*) positions that correspond to each of these (latitude, longitude) points in the projected map was obtained using standard spherical to cartesian coordinate system conversion formulas. Once this was obtained, a direct mapping of the pixel values corresponding to an *x,y,z* position in the fused image onto the projected map was performed yielding the map-projected image.

### Single-cell RNA sequencing of LPM

*Sample preparation and sorting.* Wildtype strain and *drl:mCherry*-positive zebrafish embryos were grown until tailbud stage and chorions were removed by incubating in 1 mg/mL Pronase (Roche), followed by washing in E3 medium. Around 300 embryos were used to generate the sample. The E3 medium was replaced by 1× PBS and the embryos were stored on ice until further processing. The embryos were dissociated using 2 mg/mL collagenase IV (Worthington) in DMEM (high glucose (4.5 g/L) and NaHCO₃, without L-glutamine and sodium pyruvate, Sigma-Aldrich) and incubated for 5 min in a water bath at 37 °C. The embryonic tissues were triturated into a single-cell suspension by pipetting carefully up and down. When the embryos were not yet dissociated sufficiently, they were incubated for another 5 min. Cells were filtered through a 35 μm cell-strainer (Falcon, round-bottom tubes with cell-strainer cap) and briefly centrifuged at 4427 × *g* for 30 s. Cell pellets were resuspended in 1× HBSS (Gibco) containing 2% FBS and subjected to an additional round of centrifugation and resuspension. After washing, the cells were resuspended in 1× PBS.

*drl:mCherry*-positive cells were sorted using a FACS Aria III cell sorter (BD Bioscience) and data handled using FlowJo 10.2. Cells are gated based on size and forward scattering, to exclude debris and doublets. The gating for the negative population was determined based on wildtype tailbud stage embryos. See also Supplementary Fig. 2 for gate setting. The SORTseq single-cell RNA-sequencing protocol was carried out as according to published information[65]: live mCherry-positive single cells were sorted in four 384-well plates (BioRad) containing 5 μL of CEL-Seq2 primer solution in mineral oil (24 bp polyT stretch, a 4 bp random molecular barcode (UMI), a cell-specific barcode, the 5′Illumina TruSeq small RNA kit adapter and a T7 promoter), provided by Single Cell Discoveries (Utrecht, the Netherlands). After sorting, the plates were immediately placed on ice and stored at −80 °C.

Single Cell Discoveries further processed the four plates. In brief, ERCC Spike-in RNA (0.02 μL of 1:50,000 dilution) was added to each well before cell lysis with heat shocking. Reverse transcription and second-strand synthesis reagents were dispensed using the Nanodrop II (GC biotech). After generation of cDNA from the original mRNA, all cells from one plate were pooled and the pooled sample was amplified linearly with in vitro transcription[180]. To generate sequencing libraries, RPI-series index primers were used for library PCR. Libraries were sequenced on an Illumina Nextseq500 using 75 bp paired-end sequencing.

*Single-cell analysis.* The CEL-Seq2 barcoded sequences were obtained from Single Cell Discoveries. STAR v2.5.3a[181] was used to create an index based on the Ensembl *GRCz10.91* genome and annotation, after manually adding *drl:mCherry*. Gene abundances were estimated separately for each plate using zUMIs v0.0.4[182], retaining cells with at least 100 reads and setting the Hamming distance threshold to 1 for both UMI and cell barcode identification. Finally, the exonic UMI counts for all four plates were merged into a single matrix.

Quality control and filtering were performed using the *scater* R package[183]. Upon removal of genes that were undetected across all cells, we removed cells whose percentage of mitochondrial genes fell beyond 2 Median Absolute Deviations (MADs) of the median. Secondly, features with a count greater than 1 in at least ten cells were retained for downstream analysis. Finally, we discarded cells measured on plate 4, as it was of overall low quality (low number of counts, high percentage of mitochondrial genes).

Next, we used *Seurat*[184] for clustering and dimension reduction. Clustering was performed using the 2000 most highly variable genes (HVGs) identified via *Seurat's FindVariableFeatures* function with default parameters; clustering and dimension reductions were computed using the first 20 principal components. For clustering, we considered a range of *resolution* parameters (0.2–2.4); downstream analyses were performed on cluster assignments obtained from *resolution* 1.8 (15 subpopulations).

Cluster annotations were performed manually on the basis of canonical markers in conjunction with marker genes identified programmatically with *Seurat's FindAllMarkers* function, and complementary exploration with *iSEE*[185].

For Supplementary Fig. 4, the three endoderm clusters were excluded and the remaining clusters manually merged into six major subpopulations. For each subpopulation, genes that were differentially expressed (DE) against at least three others were identified using *scater's findMarkers* function with *pval.type* = *"some"*[183]. The top 50 DE genes (in terms of effect size) at FDR < 5% and with an average positive log-fold change (*summary.logFC > 0*) were selected for visualization.

### Mesothelioma RNA-sequencing data

*Mouse model, sample preparation, and RNA-seq analysis.* Samples were derived from our previous work[128], where C57Bl/6J Nf2+/− mice were repeatedly intra-peritoneally injected with either resuspended crocidolite fibers or with saline solution (sham-exposed) every 3 weeks for eight rounds and sacrificed at 33 weeks, 12 weeks after the last crocidolite exposure (see experimental outline in Fig. 7). Tumor masses and scrapped mesothelium were collected from euthanized mice and consecutively processed for histopathological diagnosis and analysis and for RNA-seq analysis. The mice sacrificed corresponded to three groups: sham, crocidolite-exposed but no tumor, and mice exposed to crocidolite with observable tumors[128]. RNA-seq data from this previous study was used to deposited at the European Nucleotide Archive (ENA) under project accession PRJEB15230 ([http://www.ebi.ac.uk/ena/data/view/PRJEB15230]):expression values of genes associated with early LPM and mesothelium development and diagnostic markers (*Msln, Wt1, Bap1,* and *Nf2*) were extracted; counts were normalized using size factors calculated with DESeq2's estimateSizeFactors function[186]; heatmap values correspond to log10-transformed, scaled and centered (scale with default parameters) normalized counts. Right-hand side row annotations in Fig. 7 display (unscaled) log10-transformed count ranges.

*Human RNA-seq analysis.* Analysis of human mesothelioma was performed using TCGA data downloaded from cBioportal ([http://www.cBioportal.org])[187]. To perform unsupervised clustering analysis on the mRNA of TCGA mesothelioma samples, we used the *complexHeatmap* R package[188]. Heatmap values correspond to log10-transformed, batch-normalized mRNA counts.

**Reproducibility and statistics**. Statistical details of individual experiments are outlined directly in text and figures where these experiments are described. See "Methods" sections on scRNA-seq and its analysis for details on statistics. For imaging-based analysis, embryos were excluded that had aborted development following manipulation issues and technical reasons during image acquisition. Image analysis was performed on anonymized datasets. All imaging and expression analyses were performed at least twice, the majority at least three times. For the data presented in Supplementary Movie 2 and 4 individual embryos were imaged *in toto*, one dataset was fully stitched and deconvolved, and observations confirmed by the additional imaging presented in main and supplementary figures.

**Reporting summary**. Further information on research design is available in the Nature Research Reporting Summary linked to this article.

## Data availability

All sequencing datasets generated for this study have been deposited on the ArrayExpress database under accession code E-MTAB-9727. Source data for the quantification in Fig. 6G, Supplementary Fig. 8, and the heatmaps in Fig. 7 are provided with this manuscript. All reagents used and (3D) imaging datasets are freely available upon request. Raw scRNA-seq data, intermediate, and metadata files to reproduce all analyses and figures are available at [https://doi.org/10.6084/m9.figshare.13221053.v1]. The data can be explored by downloading the RDS files and using *iSEE*, an interactive user interface implemented with RStudio's *Shiny*[185]. Source data are provided with this paper.

## Code availability

All scRNA-seq analyses were run in R v4.0.2, with Bioconductor v3.11. Data preprocessing and analysis code are deposited at [https://doi.org/10.5281/zenodo.5849126], and available as a browsable *workflowr* website[189]. All package versions used throughout this study are captured in the session info provided therein. The human mesothelioma RNA-seq analyses based on TCGA data (derived from [http://www.cBioportal.org]) were run in R v4.1.0. Data preprocessing and analysis code are desposited at [https://doi.org/10.5281/zenodo.5879980]. Code of the custom-made processing steps for the mSPIM are available at [https://github.com/DaetwylerStephan/multi_sample_SPIM]. MATLAB image processing codes for the map projections of

*drl:EGFP;kop:mCherry-F-actin-nanos-3'UTR* are available at [https://github.com/sundar07/Mesothelium].

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

## Acknowledgements

We thank Single Cell Discoveries for executing the single-cell RNA-sequencing, as well as the Center for Microscopy and Image Analysis (ZMB) and the Cytometry Facility (FCF) of the University of Zurich for technical support and equipment. We also thank Dr. Deborah Yelon, Dr. Richard M. White, Dr. Charles K. Kaufman, and the members of the Mosimann lab for critical input on the manuscript. Species schematics in Figs. 1 and 7 were adjusted from templates provided by PhyloPic (phylopic.org). Zebrafish schematics in Fig. 1 were adjusted from BioRender (biorender.com). This work has been supported by Swiss National Science Foundation (SNSF) professorship PP00P3_139093, SNSF R'Equip grant 316030_150838 (Light Sheet Fluorescence Microscopy), SNSF CRSII5_180345, a Marie Curie Career Integration Grant from the European Commission (CIG PCIG14-GA-2013-631984), the Canton of Zurich, the UZH Foundation for Research in Science and the Humanities, the Swiss Heart Foundation, the ZUNIV FAN/UZH Alumni, the University of Colorado School of Medicine, Department of Pediatrics and Section of Developmental Biology, and the Helen and Arthur E. Johnson Foundation and Children's Hospital Colorado Foundation to C.M.; the Swiss Bridge Foundation to A.B. and C.M; a UZH CanDoc to S.N.; EuFishBioMed and Company of Biologists travel fellowships to K.D.P.; SNSF postdoc mobility fellowship P400PB_191057 to J.K.R.; Human Frontier Science Program (HFSP) long-term postdoctoral fellowship LT000078/2016 to S.R.N.; SNSF grant 320030_182690 and Stiftung für Angewandte Krebsforschung to E.F.B.; SNSF 310030L_182575 and ERC Starting Grant 2013 337703 to N.M.; SNSF grants 310030_175841 and CRSII5_177208 and University of Zurich's Research Priority Program Evolution in Action to M.D.R..

## Author contributions

K.D.P. and C.M. conceived the project and designed the study. K.D.P. performed the zebrafish experiments together with E.C.B. and Z.L. K.D.P., S.N., and A.K. performed and interpreted gene expression analysis in zebrafish embryos. K.D.P. and S.N. prepared and interpreted the scRNA-seq. H.L.C., C.S., and M.D.R. processed and analyzed the scRNA-seq data. S.D. and K.D.P. acquired data with multidirectional SPIM (mSPIM) and processed the data. S.R.N. generated map projection scripts. A.E. and H.S. generated *wt1b:creERT2*. D.E.C. and A.B.F. performed mouse lineage tracing experiments. M.R. performed the IHC of mouse and human tumor samples, M.R. and J.K. performed qPCR on mesothelioma human patient samples, E.F.B., M.R., and J.K. analyzed the mouse and human samples, J.K. and R.O. analyzed TCGA data. E.R., N.M., J.H., E.F.B., A.B., M.D.R., and C.M. supervised. K.D.P. compiled the data and K.D.P. and C.M. wrote the manuscript with input from all co-authors, E.F.B wrote sections of the manuscript related to mesothelioma.

## Competing interests

The authors declare no competing interests.
