## [Peer Review File · Nature Communications]

Hand2 delineates mesothelium progenitors and is reactivated in mesotheliomaREVIEWER COMMENTS

Reviewer #1 (Remarks to the Author):

The manuscript by Prummel and colleagues uses lineage tracing, transgenic fluorescent lines, and single-cell RNA sequencing in the zebrafish model to explore the origins of the mesothelium. Single cell RNA sequencing was used to good effect here to identify lateral plate mesoderm cells clusters with different expression signatures. The authors describe a domain in the lateral plate mesoderm, at the lateral-most edge, that expresses the transcription factor hand2 that delineates the progenitors of the mesothelial cells. This is verified by a combination of fate mapping using the LPM expressed gene *drl* to drive CRE in lateral plate mesoderm cells after gastrulation and further confirmed with imaging of Hand2+ cell fates using a hand2 reporter line. Complimentary fate mapping with Hand1 and Hand2 driven CRE in mice shows a contribution of cells expressing these transcription factors to the mesothelial layer in the mouse liver. This supports an evolutionary conservation of Hand+ expression in mesothelial cell progenitors. The authors show that migrating primordial germ cells cluster within the developing hand2+ mesothelial cells, and that mutants for hand2 develop defects in mesothelial tissues and fail to close their ventral body wall. Finally, the authors demonstrate that the developmental signature for mesothelial progenitors is reactivated in mesothelioma.

There is an incredible amount of data in this manuscript, but it fits together well as a story. Importantly, it is beautifully presented and convincing. I believe this will be of great interest for those interested in the mesothelium and mesothelioma, and lays the foundation for others to explore this understudied tissue more thoroughly. It is a gorgeous set of work.

To help improve readability I have the following suggestions:

1. Introduction, page 2 second paragraph – Sentence reads, “Representative for teleosts, zebrafish feature mesothelium-lined cardiac and abdominal cavities.”

I am not sure what this is supposed to help me understand. Do teleosts feature these mesothelium-lined structures and other fish species don't? Or are you saying like all teleosts, zebrafish feature these structures?

2. It would be helpful to review what the *drl* gene is and what tissues it labels and when. When does the lateral plate mesoderm form in zebrafish? Are you assured that labeling from gastrulation through 24hours will label all lateral plate mesoderm cells and nothing more?

3. Page 8 final paragraph on hand2 mutant characterization. You conclude that loss of hand2 disrupts mesothelial cell migration. How did you come to this conclusion as opposed to the idea that the progenitors failed to form or trans-fated into a different lateral plate mesoderm fate? There are GFP+ cells in the somites in your images that aren't there in the wildtype embryos for example.

Figure 1a – label the structures in the zebrafish to help those of us unfamiliar with the structures so we can understand the following images more readily. Where is the pronephros, gut, swimbladder, heart etc in the cross-section?

Figure 1C - just curiosity, but the *drl*-CRE labeled cells that project into the gut cavity are fascinating. Do you know what these are?

Figure 2G and H – these are important fluorescent in situ in terms of your story, but it is difficult to see the overlap. It would be helpful to include magnifications of the areas pointed to by the arrowheads.

Figure 3I – This diagram doesn't help me understand anything, especially without the cross-section that accompanies it in

Figure 1a. It could be removed.

Movie 3 – with the rotation, I wasn't able to see anything that was particularly informative. I'm not sure this adds much to the manuscript. What exactly do you want the reader to get from this?

Reviewer #2 (Remarks to the Author):

<ALG to author - see my comment above - I appreciate the wording is very direct here and happy to discuss by email, if you wish.>

This manuscript reports new data on the lining of the vertebrate body cavities. These are called coeloms embryonically and the tissue lining is called the mesothelium. The focus of the manuscript is on a transcription factor, *hand2*.

The initial problem with the manuscript is in the abstract. In an attempt to sell the paper they make the statement that the embryonic ontogeny and developmental regulation of the mesothelium formation remains "uncharted":. That is an almost laughable statement. Yes, the understanding of how the lateral plate mesoderm becomes subdivided into the many adult tissues that arise from this dual layer is incompletely understood – but uncharted? Hardly. In fact, they report a large number of findings on the development of the LPM. Then, in an effort to tie the mesothelium (literally the mesodermally derived epithelial lining of the body cavities) they report that in some tumors *hand2* is expressed. Any reader of this manuscript with a background in developmental biology will be turned off by this introduction. It is in the authors best interest to acknowledge that quite a bit is known about establishing the coeloms and the derivatives of the LPM and with new technologies that knowledge can be further advanced to understand how the derivative tissues diverge from the early LPM layers.

So, what is new in this manuscript and are those results sufficient to warrant publication in Nature Communicating?

In the introduction you again state: "Representative for teleosts, zebrafish feature mesothelium-lined cardiac and abdominal cavities. Which territories within the emerging LPM initially harbor the mesothelial progenitors, how and when the mesothelium diverges from other LPM lineages, and how the visceral and parietal layers form remain uncharted." This sets the stage for me. I now expect to learn whether all the LPM, both somatic and splanchnic, contribute to the mesothelium, which is what the text books indicate, or, are there subterritories only that contribute. What in this question is new?

1. In figure 1 you demonstrate what has already been reported, that *drl* is expressed in the LPM. This confirms that you built a *cre-lox* system for following *drl* expression in the territories where it is known to be expressed. This doesn't demonstrate that *drl* is expressed in all LPM cells however, and from Fig. 1D, E, and F, there is quite a bit of expression of *drl* in endodermal tissues which you suggest is "sparse labeling", even though that sparse labeling dominates the panels for liver, pancreas and pharyngeal lining. This reveals to me that your descriptions are not objective, nor do they add to what was already known about *drl* from the literature.

2. You then, at the end of this section, conclude that: "the prospective coelomic epithelium and most, if not all, developing mesothelial layers in zebrafish are LPM-derived." Now, you started this paragraph by stating that *drl*, from previous studies, is active in the LPM. And you end by stating the same thing. So you proved what you already knew, that your marker is in the LPM. You don't actually do anything advancing the notion that mesothelium is LPM derived, you simply re-confirm that finding based on earlier literature.

3. So, in answer to the first question, you don't show anything new here. The only thing that is

original is your EGFP marker expression of a gene, *drl*, that is already known to be expressed in the LPM during this time.

Next, Your paragraph in the introduction also prepares me to find out how and when the mesothelium diverges from the other LPM lineages. What in this question is new?

1. You start this, I suppose, by doing the scRNA-seq analysis and sorting the resulting cells to pick out those that co-express your *drl* marker. You show again that a number of different mesoderm and endoderm cell types co-express that marker and other endoderm or mesoderm markers.
2. Your focus switches to *hand2* because several clusters show that a high proportion of the cells in each of those several clusters express *hand2*, a gene that is well-known to be expressed in a number of LPM-derived tissues.
3. The title of this section begins with "The early LPM consists of distinct progenitor populations". Why do you state this? What you are actually observing is that once again you are showing that *drl* is expressed in several mesodermal and endodermal populations, something you've already established. And since we already know that the LPM contains cells that will become a variety of tissues this isn't new. Rather, you add a new resource to what is already established by showing cell clusters that can add unknown genes to the already known genes in a given cluster. This could have some utility. However, since it is a snapshot in time of development some of the clusters could also reflect several stages of the same cell type and this becomes problematic with your several *hand2* clusters.
4. You add some expression studies of *Wt1*, which as previously reported, is in LPM, and you then conclude, based on *hand2* co-expression, that *hand2* is an earlier and more complete marker of the LPM. In going back to the title of this section, yes, *hand2* is a marker in the LPM and is also later expressed in the mesothelium. What I don't see is the answer to the how and when the mesothelium diverges from the other LPM lineages.
5. I really like your Figure 4 showing PGCs in association with the *drl*-expressing LPM and the *hand2*-expressing LPM. Your conclusion, however, puzzles me since it was already known that PGCs travel along the coelomic epithelium and up the dorsal mesentery, where it exists, to reach the genital ridges. Yes, you show that *drl* and *hand2* are expressed in those cells at that time, but it still doesn't help me understand how the mesothelium differentiates from the LPM.
6. The data on the time lapse imaging is also very nice as it shows some of the dynamics of peritoneum formation as seen with the *hand2* marker. Further, the perturbation experiment with the *hand2*-null shows that its function is necessary for the migratory activity involved in establishing the mesothelial lining.

Finally, that paragraph in the introduction prepares me to find out how the visceral and parietal layers form. What in this question is new?

1. I've gone over the manuscript several times and I'm at a loss to learn how the expression of two markers, *drl* and *hand2* add new information to address this question. Yes, both are markers but that doesn't really address the How. *Hand2* is present during the migratory activity implying that it is somehow functioning in cells, and the perturbation experiment supports that but that still doesn't tell me how the two layers form other than *hand2* being necessary for the migratory activity. The movies show the dynamics of the event very nicely but we knew the cells migrated already.

In sum, this reviewer finds that there are some nice data on expression of *hand2* as a marker of the coelomic epithelium and expression of a GFP marker delineating *drl* over time, but the claims of enlightening us of an uncharted developmental sequence is vastly overstated. And, as one reads the manuscript, there are overstatements throughout for a manuscript that simply documents expression of two genes in the development of the LPM.

Reviewer #3 (Remarks to the Author):

Review: NCOMMS-20-47883-T

Hand2 delineates mesothelium progenitors and is reactivated in mesothelioma

To me this manuscript can be divided into two parts. The first part is directed at the developmental biology of the mesothelium. The authors describe many experiments focusing on the very early origin of mesothelium and its involvement in establishing the architecture and homeostasis of the internal organs in the zebrafish. The use several sophisticated and elegant procedures including genetic lineage tracing, in toto live imaging and single-cell transcriptomics, in zebrafish to track the origin of mesothelial progenitors from the lateral plate mesoderm (LPM). Based on single cell analysis they uncover the transcription factor Hand2 as central in driving distinct progenitor populations within the forming LPM. Next, they show that the function of Hand2 in zebrafish has been evolutionary conserved in the mouse. Loss of function Hand2 mutants leads to similar defects in zebrafish and mice, e.g. aberrant development of heart, limbs and smooth muscle layers. More specifically, in zebrafish they show that loss of Hand2 leads to disturbed mesothelial progenitor migration and effectively to malformation of mesothelial lining. Most of this part of the manuscript might be rather descriptive, but it gives a wealth of elegant data illustrating the significance of Hand2 in the control of mesothelium development and on the organization of the pericardium and visceral and parietal peritoneum.

In the second part, inspired by its critical role in mesothelium development the authors of investigate its significance for mesothelioma formation. Most human mesothelioma cases are linked to asbestos fibers exposure. The authors start using crocidolite (blue asbestos) exposure to induce mesotheliomas in the Nf2 mouse model. Histological and transcriptional analyses of the induced mesotheliomas strongly suggest a role in their formation of Hand2 and many of its transcriptional targets, also implicated in the setup of the mesothelial lining in both zebrafish and mice. Next, they turned to a large publicly available data set of human mesotheliomas to confirm this suggestion. In contrast to the panel of reproducible Nf2 mouse tumors, this data set is based on a large variety of mesotheliomas which show extensive differences in expression patterns and likely also in mutational spectrum. Therefore it might not come as a surprise that this expression analysis does not warrant any suggestion on the role of Hand2 mesothelioma formation in humans. In addition, the latency of human mesothelioma development often takes many years after initial asbestos exposure in contrast to the development in mice, which by itself might add to very broad tumor heterogeneity complicating data interpretation. Nevertheless, the data human mesothelioma presented are confusing and somehow disappointing. A suggestion would be to stratify the mesothelioma cases on the bases of genetic mutations, e.g. loss of NF2, BAP1 and maybe CDKN2ab, and histological subtype. Also, other, larger, mesothelioma data sets could also be used.

A few questions/remarks:

- 1) Related to the mouse mesothelioma experiment: is the wt Nf1 allele lost in the tumors? What are the histological subtypes of mesotheliomas
- 2) Nf2, Hand2 and Bap1 immunohistochemistry of control and crocidolite exposed mesothelium would be very informative.
- 3) It would be very informative to provide data on the subtype of the human mesotheliomas and their genetic mutational spectrum.

REVIEWER COMMENTS

Reviewer #1 (Remarks to the Author):

The manuscript by Prummel and colleagues uses lineage tracing, transgenic fluorescent lines, and single-cell RNA sequencing in the zebrafish model to explore the origins of the mesothelium. Single cell RNA sequencing was used to good effect here to identify lateral plate mesoderm cells clusters with different expression signatures. The authors describe a domain in the lateral plate mesoderm, at the lateral-most edge, that expresses the transcription factor *hand2* that delineates the progenitors of the mesothelial cells. This is verified by a combination of fate mapping using the LPM expressed gene *drl* to drive CRE in lateral plate mesoderm cells after gastrulation and further confirmed with imaging of *Hand2*⁺ cell fates using a *hand2* reporter line. Complimentary fate mapping with *Hand1* and *Hand2* driven CRE in mice shows a contribution of cells expressing these transcription factors to the mesothelial layer in the mouse liver. This supports an evolutionary conservation of *Hand*⁺ expression in mesothelial cell progenitors. The authors show that migrating primordial germ cells cluster within the developing *hand2*⁺ mesothelial cells, and that mutants for *hand2* develop defects in mesothelial tissues and fail to close their ventral body wall. Finally, the authors demonstrate that the developmental signature for mesothelial progenitors is reactivated in mesothelioma.

There is an incredible amount of data in this manuscript, but it fits together well as a story. Importantly, it is beautifully presented and convincing. I believe this will be of great interest for those interested in the mesothelium and mesothelioma, and lays the foundation for others to explore this understudied tissue more thoroughly. It is a gorgeous set of work.

To help improve readability I have the following suggestions:

We much appreciate the reviewer's enthusiastic take on our work as well as the constructive input in the comments. We also hope our data will inform and motivate researchers to tackle the early origins of the mesothelium also beyond zebrafish!

1. Introduction, page 2 second paragraph – Sentence reads, “Representative for teleosts, zebrafish feature mesothelium-lined cardiac and abdominal cavities.” I am not sure what this is supposed to help me understand. Do teleosts feature these mesothelium-lined structures and other fish species don't? Or are you saying like all teleosts, zebrafish feature these structures?

We indeed aimed to introduce zebrafish as example for teleosts that feature a homologous, yet still distinct series of mesothelium-lined body cavities to terrestrial vertebrates as teleosts lack lungs. We now rephrased the sentence to avoid any confusion with future readers of our work (“Despite teleosts such as zebrafish lacking the pleural cavity due to lack of lungs, they feature mesothelium-lined cardiac and abdominal cavities.”)

2. It would be helpful to review what the *drl* gene is and what tissues it labels and when. When does the lateral plate mesoderm form in zebrafish? Are you assured that labeling from gastrulation through 24hours will label all lateral plate mesoderm cells and nothing more?

This is a critical point that we have, in retrospect, oversimplified in our initial submission. We and others had previously published the details of the tissue labeling resulting from *drl:creERT2*-based lineage tracing (Prummel et al., 2019; Sagarin et al., 2019; Felker et al., 2018; Henninger et al., 2017; Gays et al., 2017; McCarthy et al., 2016; Mosimann et al., 2015) and we wanted to avoid repetition to emphasize the new observations we made in the work presented here. Nonetheless, the reviewer's comment made us aware that we should not expect/force any future reader to first read our previous work in detail as basis for this manuscript – we have now added further explanation in the text and the corresponding Figure 1 with added LPM schematics for context to better emphasize as to what is known and what the novel aspects are of our work (also in response to reviewer 2).

From our previous use of *drl*-based reporters and also reports of other labs applying our reagents, we are confident that these transgenics are reporting LPM-primed mesendoderm from gastrulation stages and become increasingly restricted to all known LPM lineages (or those known to date and the mesothelia reported here). Our data presented here using scRNA-seq further corroborate this conclusion in our documentation that *hand2*-expressing posterior LPM forms mesothelia.

3. Page 8 final paragraph on *hand2* mutant characterization. You conclude that loss of *hand2* disrupts mesothelial cell migration. How did you come to this conclusion as opposed to the idea that the progenitors failed to form or trans-fated into a different lateral plate mesoderm fate? There are GFP⁺ cells in the somites in your images that aren't there in the wildtype embryos for example.

We took a closer look at the *hand2* mutants and have performed live light sheet-based 3D imaging of several wildtype and *hand2* mutant in *hand2:EGFP* background, allowing us to in more detail observe how the mesothelial progenitors

behave. We observe in these 3D renderings that the majority of the *hand2:EGFP* cells fail to properly migrate ventrally over the yolk and yolk extension, except of a few scattered cells. The rare GFP-expressing cells within the paraxial mesoderm could be incorrectly migrated LPM cells. Besides expanding the figure with lateral views of the *hand2* mutants, we also added Movie 4, providing three-dimensional view of the disrupted migration.

Figure 1a – label the structures in the zebrafish to help those of us unfamiliar with the structures so we can understand the following images more readily. Where is the pronephros, gut, swimbladder, heart etc in the cross-section?

We thank the reviewer for this suggestion. We have refined the labeling in the figures and hope that the images are better to understand for an untrained eye. Additionally, we included cartoons of the LPM to the schematic of our lineage tracing workflow, intending to familiarize the reader already in Figure 1 with the architecture of the LPM during early development. We hope that the imaging of the bilateral LPM in Figure 3 will be easier to follow.

Figure 1C - just curiosity, but the *drl*-CRE labeled cells that project into the gut cavity are fascinating. Do you know what these are?

The reviewer has a keen eye! These individual cells pointing into the gut at the here chosen 4-OHT induction time points are endoderm-derived cells that contribute to the gut lining, representing the occasional endoderm cell deriving from the *drl*-expressing progenitors. We previously documented that endoderm vs LPM labeling follows a temporal (i.e. the later induced, the more selective LPM labeling) and a spatial (i.e. posterior *drl*-expressing cells transition later to sole LPM fate) pattern (Prummel et al., 2019 and Gays et al., 2017).

Figure 2G and H – these are important fluorescent in situs in terms of your story, but it is difficult to see the overlap. It would be helpful to include magnifications of the areas pointed to by the arrowheads.

We have included a zoom-in of the interesting regions with overlap.

Figure 3I – This diagram doesn't help me understand anything, especially without the cross-section that accompanies it in Figure 1a. It could be removed.

We agree with the reviewer that the diagram in Fig 3I does not add anything to the figure and we have removed it.

Movie 3 – with the rotation, I wasn't able to see anything that was particularly informative. I'm not sure this adds much to the manuscript. What exactly do you want the reader to get from this?

We realized that Movie 3 went too far in the psychedelic spectrum and we have replaced it with a simpler split-screen video showing a dorsal view, a lateral view, and a more rostral view of the forming peri- and epicardium. We hope this video is more pleasing to the readers and supports our figures and text. We additionally added Movie 4 to provide a 3D impression of *hand2:EGFP* in wildtype vs *hanS6* mutants towards further emphasizing the migration defect in the mutants.

Reviewer #2 (Remarks to the Author):

This manuscript reports new data on the lining of the vertebrate body cavities. These are called coeloms embryonically and the tissue lining is called the mesothelium. The focus of the manuscript is on a transcription factor, *hand2*.

The initial problem with the manuscript is in the abstract. In an attempt to sell the paper they make the statement that the embryonic ontogeny and developmental regulation of the mesothelium formation remains “uncharted”. That is an almost laughable statement. Yes, the understanding of how the lateral plate mesoderm becomes subdivided into the many adult tissues that arise from this dual layer is incompletely understood – but uncharted? Hardly. In fact, they report a large number of findings on the development of the LPM. Then, in an effort to tie the mesothelium (literally the mesodermally derived epithelial lining of the body cavities) they report that in some tumors *hand2* is expressed. Any reader of this manuscript with a background in developmental biology will be turned off by this introduction. It is in the authors best interest to acknowledge that quite a bit is known about establishing the coeloms and the derivatives of the LPM and with new technologies that knowledge can be further advanced to understand how the derivative tissues diverge from the early LPM layers.

So, what is new in this manuscript and are those results sufficient to warrant publication in Nature Communicating?

In the introduction you again state: “Representative for teleosts, zebrafish feature mesothelium-lined cardiac and abdominal cavities. Which territories within the emerging LPM initially harbor the mesothelial progenitors, how and when the mesothelium diverges from other LPM lineages, and how the visceral and parietal layers form remain uncharted.” This sets the stage for me. I now expect to learn whether all the LPM, both somatic and splanchnic, contribute to the mesothelium, which is what the text books indicate, or, are there subterritories only that contribute. What in this question is new?

1. In figure 1 you demonstrate what has already been reported, that *drl* is expressed in the LPM. This confirms that you built a *cre-lox* system for following *drl* expression in the territories where it is known to be expressed. This doesn't demonstrate that *drl* is expressed in all LPM cells however, and from Fig. 1D, E, and F, there is quite a bit of expression of *drl* in endodermal tissues which you suggest is “sparse labeling”, even though that sparse labeling dominates the panels for liver, pancreas and pharyngeal lining. This reveals to me that your descriptions are not objective, nor do they add to what was already known about *drl* from the literature.

As also reviewer 1 pointed out, we have been rather sparse in re-introducing previous work that we and others have performed using *drl*-based transgenic reporters in zebrafish and other chordate models. We have deliberately aimed to avoid writing overly repetitive accounts of previous work that we mainly cited in the text – based on the reviewers' input, we have now added further context and background to our used tools to aid the uninitiated reader through the individual steps and findings. We further point out, to the best of our knowledge, what we thought were the gaps in the literature before we started this work.

In several previous manuscripts from our group, we outlined in detail that gastrulation-stage *drl:EGFP*, *drl:mCherry*, and *drl:creERT2*-mediated lineage tracing labels LPM-primed mesendoderm, resulting in lineage labeling across all known LPM derivatives as well as endodermal cells (Mosimann et al., 2015; Gays et al., 2017; Henninger et al., 2017; Felker et al., 2018; Sanchez-Iranzo et al., 2018; Prummel et al., 2019; Pfefferli et al., 2021). The endodermal labeling becomes increasingly less during early development as the *drl* regulatory elements (specifically the intronic pan-LPM enhancer) becomes specific to the LPM (Prummel et al., 2019). The developmental nature of the LPM as part of the mesendoderm is therefore still visible in the carry-over of endodermal labeling, as also validated in other model systems (Prummel et al., 2019, done in several chordate models by experts in their respective field).

Given this wealth of previous data that our work presented here builds on, we frankly fail to see how our descriptions can be deemed “not objective”. Further, the reviewer states further below that mesothelial origins from LPM are “nothing new” to the reviewer – which would further validate our experimental findings presented in Figure 1.

Overall, Figure 1 sets the stage that we can use our transgenic means to find the earliest mesothelial progenitors as part of the emerging LPM, validating our then chosen scRNA-seq approach based on sorting of *drl:mCherry*.

2. You then, at the end of this section, conclude that: “the prospective coelomic epithelium and most, if not all, developing mesothelial layers in zebrafish are LPM-derived.” Now, you started this paragraph by stating that *drl*, from previous studies, is active in the LPM. And you end by stating the same thing. So you proved what you already knew, that your marker is in the LPM. You don't actually do anything advancing the notion that mesothelium is LPM derived, you simply re-confirm that finding based on earlier literature.

3. So, in answer to the first question, you don't show anything new here. The only thing that is original is your EGFP marker expression of a gene, *drl*, that is already known to be expressed in the LPM during this time.

Addressing points 2 and 3:

We have rephrased and clarified the main text as a response to the reviewer's comment (p. 4 end of paragraph). The reviewer presents our Figure 1 data as it is a circular argument, an interpretation with which we respectfully disagree. We

have previously established pan-LPM marking by *drl*-based transgenics as first such tool across vertebrate models. We have emphasized that the transgenic zebrafish strains used are the tools allowing us to study the prospective coelomic epithelium and the developing mesothelial layers. Our data presented throughout our manuscript establishes that *drl:EGFP*, *drl:mCherry*, and *drl:creERT2* transgenics are suitable to further dive into the cells-of-origin of the mesothelium within the LPM. Figure 1 provides the conceptual intro for the use of our subsequent tools to uncover the earliest signature of mesothelial progenitors within the emerging LPM and linking these to early *hand2* expression beyond cardiac and cardiopharyngeal progenitors.

Next, Your paragraph in the introduction also prepares me to find out how and when the mesothelium diverges from the other LPM lineages. What in this question is new?

We are not sure we understand the reviewer's criticism. While the question might not be "new", it is to our knowledge still unresolved how the mesothelial progenitors are set apart from the rest of the LPM-derived cell fates. Especially in the era of single-cell analysis where mesothelium and other "mesenchymal" cell types are increasingly appearing in various experimental systems (but often poorly described), a better understanding of the embryonic progenitors of these cell types will be highly instrumental to the field. In addition, to understand the cell-of-origin of cancers like mesothelioma, more insights into key stem and developmental genes are needed.

1. You start this, I suppose, by doing the scRNA-seq analysis and sorting the resulting cells to pick out those that co-express your *drl* marker. You show again that a number of different mesoderm and endoderm cell types co-express that marker and other endoderm or mesoderm markers.

2. Your focus switches to *hand2* because several clusters show that a high proportion of the cells in each of those several clusters express *hand2*, a gene that is well-known to be expressed in a number of LPM-derived tissues.

Addressing point 1 and 2 raised above by the reviewer:

We have performed single cell RNA sequencing on *drl:mCherry*-expressing cells at tailbud stage. We have dissociated the embryos into a single-cell suspension and sorted the cells endogenously expressing mCherry, representing the LPM (and a few endoderm) cells at tailbud stage. This provided us a first targeted dataset with a high resolution of the LPM progenitors and towards identifying the complement of potential cell fate origins within this mesoderm compartment. Our finding of the endodermal clusters is in line with our previous imaging and lineage tracing studies of the *drl*-expressing cells (Prummel et al., 2019), where we documented a gradual specification of the mesendoderm to LPM. Our transcriptome data is also in line with previously published zebrafish scRNA-seq of whole embryos (Farrell et al., 2018; Wagner et al., 2018; Farnsworth et al., 2020; Song et al., 2020), however, our approach provides a resolution of the progenitor fields within the LPM with a yet undescribed detail.

Using established marker genes, we carefully identified various progenitor fields, among which the heart field and kidney and endothelium progenitors. Additionally, our analysis also revealed several clusters of an unclear identity. The common denominator for these clusters is the expression of *hand2*, a transcription factor gene that has indeed been previously described and extensively studied in cardiac and limb development. Nevertheless, we observed *hand2* expression in clusters beyond the cardiac progenitors, as outlined in Figure 2 with context of co-expressed genes. From the complete expression patterns and overlaps, as well as previous work that left the cellular identity of especially posterior LPM-residing *hand2*-expressing cells uncharacterized (i.e. the *cdx4*-positive, *hand2*-expressing LPM in our dataset), we then expanded our analysis with live imaging and transgenic readouts to establish that these mystery cells indeed represent mesothelial progenitors.

Consequently, the *hand2*-expressing early LPM that is not cardiac, pharyngeal, or forelimb-fated, represents mesothelial progenitors, which to our knowledge is the first gene-expression based evidence for these progenitors at these early developmental stages.

3. The title of this section begins with "The early LPM consists of distinct progenitor populations". Why do you state this? What you are actually observing is that once again you are showing that *drl* is expressed in several mesodermal and endodermal populations, something you've already established. And since we already know that the LPM contains cells that will become a variety of tissues this isn't new. Rather, you add a new resource to what is already established by showing cell clusters that can add unknown genes to the already known genes in a given cluster. This could have some utility. However, since it is a snapshot in time of development some of the clusters could also reflect several stages of the same cell type and this becomes problematic with your several *hand2* clusters.

We do agree with the referee that the section title doesn't tickle the reader with a sensational headline – indeed, we use the title chosen to guide the reader to the next part of the manuscript, in which we outline the performed scRNA-seq experiment (which to our knowledge is the first of its kind on early LPM and in resolving the different progenitor populations present at the end of gastrulation already). We have now rephrased the section title to "The early LPM harbors distinct *hand2*-expressing cell clusters".

We are aware that distinct clusters can represent several stages of the same cell type and have added this phrasing to the main text (p. 6 at the conclusion of the scRNA-seq section; p. 12 in the discussion when revisiting the genes co-expressed with *hand2*), however, we respectfully disagree that this would be problematic with our several *hand2* clusters. First, *hand2* has been well-established as a cardiac and pharyngeal transcription factor gene, as three clusters in our scRNA-seq recapitulate (“heart field” and “cardiopharyngeal”), and as acknowledged by our co-authors who have been working with *Hand2* since decades in mouse and beyond. Second, we additionally find 4 *hand2*-high clusters that are neither of cardiac, nor of pharyngeal identity. Some of these clusters represent cells that are clearly posterior LPM as per co-expression of the unambiguous posterior gene *cdx4* (Fig. 2 and following), where others represent anterior, but not endo-of myocardial progenitor populations. The identity of these cells, their cell state at the moment of transcriptome isolation, or their final fates are not evident from our scRNA-seq. Nonetheless, our analysis reveals that several genes previously linked to bilateral LPM expression, in particular *foxh1*, are expressed in the lateral-most LPM that co-expresses *hand2* and these cells are indeed of so-far unassigned fate (see also Osterwalder et al., 2014; Gays et al., 2017; Perens et al., 2016; Perens et al., 2020; reviewed in our recent review Prummel et al., 2020).

Our accumulated evidence outlined in the subsequent figures and movies let us conclude that the non-cardiac/pharyngeal clusters in our scRNA-seq represent mesothelial progenitors. To our knowledge and building upon work we amply cited in our manuscript that links the mesothelium to an LPM origin, our entire study is the first report linking early *hand2* expression to mesothelial progenitors, significantly adding to the field’s previous focus on *hand2* as cardiac and pharyngeal gene – as the reviewer also acknowledges in their next comment below.

4. You add some expression studies of *Wt1*, which as previously reported, is in LPM, and you then conclude, based on *hand2* co-expression, that *hand2* is an earlier and more complete marker of the LPM. In going back to the title of this section, yes, *hand2* is a marker in the LPM and is also later expressed in the mesothelium. What I don’t see is the answer to the how and when the mesothelium diverges from the other LPM lineages.

Similar to the reviewer’s comment above, we do not claim that we provide any definitive answer to how and when mesothelium diverges from the other LPM lineages. However, our manuscript provides critical data towards addressing these still open questions. We push our knowledge to the end of gastrulation (and possibly to slightly before) with uncovering that a subset of *hand2*-expressing LPM progenitors are already primed to become mesothelium. As outlined also above, we have rephrased the text passages to more precisely outline what our new work reveals (and what remain unanswered questions or mechanisms).

Throughout our manuscript, however, we do provide evidence that the divergence happens already at the end of gastrulation, i.e. through our scRNA-seq, the live-imaging of the *hand2:EGFP* reporter, and as referenced to an extensive body of previous work that characterized the onset and specificity of *Hand2* gene expression in different vertebrates throughout development (in the lateral-most LPM that has so-far been unclear as to what it becomes). Additionally, we also make clear that *hand2* is not a more complete marker of the LPM, as it only is expressed in a subset of the LPM (in contrary to some other statements in literature), and its expression is restricted to the lateral-most LPM and cardiopharyngeal field. The *hand2* expression within the *drl*-expressing LPM segregates the blood, vasculature, and kidney progenitors from the *hand2*-expressing LPM.

Our expression studies for the zebrafish *wt1* genes, which are the closest to a broad mesothelial marker across vertebrates to date, were meant to support our claims that *hand2* expression marks mesothelia and to link our findings to what has previously been used in the field. Our work uses validated transgenic reporters and provide coverage for their expression throughout the selected timepoints, further supporting our conclusion that a) *wt1* genes in zebrafish provide mesothelial markers for select aspects of mesothelial regions, and b) *hand2* is active before the onset of *wt1a* and *wt1b* expression. These points connect to previous work and further substantiate that *hand2* is a new and to our knowledge the earliest mesothelial marker in development.

To further validate this point, we have now added further light sheet-based 3D imaging of *hand2:EGFP* in wildtype and in *hand2* mutants as well as 3D renderings in video form that show the extent of parietal (and, as resolution allows, visceral) mesothelium cells covering the zebrafish embryo. These renderings further reveal how mesothelial progenitors fail to properly migrate in *hand2* mutants, adding an early developmental phenotype to the known repertoire of mesothelium-perturbing factors reported previously.

5. I really like your Figure 4 showing PGCs in association with the *drl*-expressing LPM and the *hand2*-expressing LPM. Your conclusion, however, puzzles me since it was already known that PGCs travel along the coelomic epithelium and up the dorsal mesentery, where it exists, to reach the genital ridges. Yes, you show that *drl* and *hand2* are expressed in those cells at that time, but it still doesn’t help me understand how the mesothelium differentiates from the LPM.

The referee is correct in stating that the position of the migrating PGCs was previously studied at similar stages we present in this work, i.e. including gastrulation stages and early somitogenesis. However, the analysis we present here is novel with respect to the level of resolution. Specifically, the location of the PGCs has not been studied relative to the now-visualized mesothelial cell layer that is labelled with the *hand2:EGFP* reporter, and this allowed us to clearly show the tight association between the two cell populations and relative to the more dorsal kidney progenitors (the migration boundary visualized in

previous work and the usual standard in the field). This information will be of importance for the ongoing and future study of the mechanisms that position the PGCs, for example supporting the idea that cell-cell interactions are important.

6. The data on the time lapse imaging is also very nice as it shows some of the dynamics of peritoneum formation as seen with the *hand2* marker. Further, the perturbation experiment with the *hand2*-null shows that its function is necessary for the migratory activity involved in establishing the mesothelial lining.

We are grateful to hear the reviewer's appreciation of our time lapse imaging and the observations with the *hand2* mutants. We hope that our light sheet imaging of the emerging pericardium and peritoneum will bring this tissue to the awareness of (zebrafish) researchers when performing live imaging of similar time points and transgenic zebrafish strains. This will contribute to gathering more data on this fascinating tissue in the future.

Finally, that paragraph in the introduction prepares me to find out how the visceral and parietal layers form. What in this question is new?

1. I've gone over the manuscript several times and I'm at a loss to learn how the expression of two markers, *drl* and *hand2* add new information to address this question. Yes, both are markers but that doesn't really address the How. *Hand2* is present during the migratory activity implying that it is somehow functioning in cells, and the perturbation experiment supports that but that still doesn't tell me how the two layers form other than *hand2* being necessary for the migratory activity. The movies show the dynamics of the event very nicely but we knew the cells migrated already.

As now also further clarified in our edited text, our work does not claim nor aim to resolve all mesothelial development. Instead, our work provides: 1) the first developmental evidence for the mesothelial progenitor identity at the end of gastrulation through *hand2* expression in zebrafish, 2) a rudimentary mesothelial progenitor program in the *hand2*-expressing mesothelial progenitors, 3) imaging of the mesothelium formation in a vertebrate embryo as per our movies; 4) evidence that links *hand2*'s function to mesothelium migration; and 5) indications for reactivated expression of *Hand2* and other early LPM/mesothelial progenitor genes in mesothelioma tumors. We strived to place all our data in context of what is already known in this fascinating field. Our work allows for future studies to for example further unravel the molecular mechanisms (i.e. the function of genes co-expressed with *hand2*) involved in the segregation between the visceral and parietal layers, in particular with a focus on their migration towards the embryo midline (visceral mesothelium) and in the lateral-dorsal direction (parietal mesothelium).

In sum, this reviewer finds that there are some nice data on expression of *hand2* as a marker of the coelomic epithelium and expression of a GFP marker delineating *drl* over time, but the claims of enlightening us of an uncharted developmental sequence is vastly overstated. And, as one reads the manuscript, there are overstatements throughout for a manuscript that simply documents expression of two genes in the development of the LPM.

As outlined in our response and as also summarized by reviewer 1, our work provides the first link of early *hand2* expression to mesothelial progenitor identity within the emerging LPM. To our knowledge, the earliest stages of mesothelial progenitor emergence had never been defined at the gene expression level and in their context within the emerging LPM, all information that our study now provides. We further emphasize how the emerging mesothelium develops in the context of the overall LPM diversification by linking to known phenomena in other vertebrates (i.e. *wt1* gene expression), how the early coelomic epithelium interacts with and provides context for other developmental processes (i.e. PGC migration), and how a previously unanticipated function of *Hand2* contributes to mesothelium migration and integrity. We sincerely apologize that our initial text was interpreted differently by the reviewer, and we strived to clarify, streamline, and further support our manuscript with the here presented revised version (and responses to the fellow reviewers 1 and 3). We hope that our work provides new impulses to the study of the often neglected mesothelium and are looking much forward to learning more about its still mysterious biology.

Reviewer #3 (Remarks to the Author):

To me this manuscript can be divided into two parts. The first part is directed at the developmental biology of the mesothelium. The authors describe many experiments focusing on the very early origin of mesothelium and its involvement in establishing the architecture and homeostasis of the internal organs in the zebrafish. The use several sophisticated and elegant procedures including genetic lineage tracing, in toto live imaging and single-cell transcriptomics, in zebrafish to track the origin of mesothelial progenitors from the lateral plate mesoderm (LPM). Based on single cell analysis they uncover the transcription factor Hand2 as central in driving distinct progenitor populations within the forming LPM. Next, they show that the function of Hand2 in zebrafish has been evolutionary conserved in the mouse. Loss of function Hand2 mutants leads to similar defects in zebrafish and mice, e.g. aberrant development of heart, limbs and smooth muscle layers. More specifically, in zebrafish they show that loss of Hand2 leads to disturbed mesothelial progenitor migration and effectively to malformation of mesothelial lining. Most of this part of the manuscript might be rather descriptive, but it gives a wealth of elegant data illustrating the significance of Hand2 in the control of mesothelium development and on the organization of the pericardium and visceral and parietal peritoneum.

In the second part, inspired by its critical role in mesothelium development the authors of investigate its significance for mesothelioma formation. Most human mesothelioma cases are linked to asbestos fibers exposure. The authors start using crocidolite (blue asbestos) exposure to induce mesotheliomas in the Nf2 mouse model. Histological and transcriptional analyses of the induced mesotheliomas strongly suggest a role in their formation of Hand2 and many of its transcriptional targets, also implicated in the setup of the mesothelial lining in both zebrafish and mice. Next, they turned to a large publicly available data set of human mesotheliomas to confirm this suggestion. In contrast to the panel of reproducible Nf2 mouse tumors, this data set is based on a large variety of mesotheliomas which show extensive differences in expression patterns and likely also in mutational spectrum. Therefore it might not come as a surprise that this expression analysis does not warrant any suggestion on the role of Hand2 mesothelioma formation in humans. In addition, the latency of human mesothelioma development often takes many years after initial asbestos exposure in contrast to the development in mice, which by itself might add to very broad tumor heterogeneity complicating data interpretation. Nevertheless, the data human mesothelioma presented are confusing and somehow disappointing. A suggestion would be to stratify the mesothelioma cases on the bases of genetic mutations, e.g. loss of NF2, BAP1 and maybe CDKN2ab, and histological subtype. Also, other, larger, mesothelioma data sets could also be used.

We much appreciate the reviewer's nuanced take on our manuscript's developmental and disease-related aspects of the mesothelium. While we do agree that the descriptive aspects of our work (Figures 1-5) are the bulk of the presented work as new framework for the context of earliest mesothelial progenitor formation, the linking of Hand2 function to proper mesothelium formation adds a molecular player to the process (Figure 6). We have now added further imaging evidence for *hand2* mutant phenotypes affecting the mesothelium, underlining the previously unanticipated and unappreciated role of this gene/transcription factor in mesothelium formation.

Building on these advances, the connection of *Hand2* expression to mouse and human mesothelioma is by far not of the same weight as our other presented data, i.e. represented as Figure 7. In its intended form, our work aims to provide a first documentation and linkage of Hand2 expression and function to mesothelial biology in development and disease. Our work was not intended to tackle mesothelioma as main focus. Yet, the association of Hand2 and other LPM-related genes with at least a subset of the human mesothelioma cases provides a significant step towards in stratifying tumors and potentially for mechanistic follow-up studies to understand what makes these mesothelioma subsets distinct. Together, our manuscript documents the earliest context of mesothelial progenitor formation within the LPM, and links (previously unknown) features of mesothelial progenitor biology in development to mesothelioma. We hope our work and results inspire additional labs in the field to build on our findings.

In our revised manuscript, we have taken the reviewer's input to heart and added the new Supplementary Figure 8 to further provide context to Figure 7's tumor data based on the wealth of info from the TCGA database. We outline this and the other edits and additions below.

A few questions/remarks:

1) Related to the mouse mesothelioma experiment: is the wt Nf1 allele lost in the tumors? What are the histological subtypes of mesotheliomas

We appreciate and share the reviewer's interest in *Nf2* status. We had documented the loss of the remaining *Nf2* allele in one mesothelioma line derived from one of the tumors in our previous work (Rehrauer et al, Oncogene 2018). In the same manuscript, we had also documented that the mouse tumors had spindle morphology.

2) Nf2, Hand2 and Bap1 immunohistochemistry of control and crocidolite exposed mesothelium would be very informative.

Nf2: to our knowledge, Nf2 expression has been documented in mice tissue using *in situ* hybridization (gene expression) (e.g. McClatchey et al, Genes and Development 1997) or by Western blot analysis of tumor lysates (protein expression) (e.g. Kukuyan et al, Cancer Research 2019). There are, to our knowledge, no commercial antibodies against Nf2 that are compatible with immunohistochemistry. Concerning Bap1, to our knowledge, there is only one antibody sufficiently working for IHC in mice, which has been developed by Anwasha Dey, Genentech (Gu et al, Cancer Discovery 2017). We have spent considerable time and effort to obtain this reagent without success. However, other work has documented Bap1 expression by Western blot analysis of tumor lysates (e.g. Kukuyan et al, Cancer Research 2019). In Rehrauer et al Oncogene 2018, we documented the loss of one Bap1 allele in the same mesothelioma line mentioned above, which was derived from one of the tumors.

Concerning Hand2, our sequencing data establishes that Hand2 expression in wildtype/sham-treated mouse mesothelium tissue is low to undetectable (Fig. 7), correlating with previous work on Hand2 reporters and mRNA *in situ* hybridization (including work from co-authors, i.e. Barnes et al., 2011; Firulli et al., 1998; Han and Olson, 2005; Shin et al., 2009). We have now added to Supplementary Figure 8 the figure panel B, depicting sections of crocidolite-exposed mouse mesothelium stained with H&E, antibody staining control, and anti-Hand2. As the section reveals, Hand2 expression appears in a mosaic fashion throughout the exposed mesothelium, in line with our sequencing results and with our model that asbestos exposure (and likely its associated inflammation) leads to a increasing re-activation of early LPM factors active in mesothelial progenitors, in particular of Hand2. We feel these figure panels are a valuable new addition to the manuscript based on the reviewer's excellent input.

Taken together, we believe this cumulative data provides sufficient context for our finding of re-activated LPM gene expression in mesothelioma, and in particular of *Hand2*, as extension of our overarching work to document the earliest emergence of mesothelial progenitors in development.

3) It would be very informative to provide data on the subtype of the human mesotheliomas and their genetic mutational spectrum.

This is an excellent suggestion by the reviewer. We have now added Supplementary Figure 8 and Supplementary Table 1 to the manuscript, in which we provide TCGA data-based stratification of the analyzed mesothelioma tumors with respect to the *CDKN2A* and *BAP1* mutational spectrum. These two genes were selected on the basis that they are the most abundantly mutated genes in mesothelioma. The new supplementary figure and table are meant to complement our heatmap in Figure 7, and Supplementary Table 1 also contains *NF2* data for completion. Altogether, our data further underline that a subset of mesothelioma that does not stratify by a particular mutational composition of *CDKN2A* and *BAP1*, re-activates early LPM/mesothelial progenitor genes, in particular Hand2 and Meis factors. We are excited about the possibility that mesothelioma researchers could correlate this LPM signature with their datasets towards gaining more insights into how this deadly tumor operates.

REVIEWER COMMENTS

Reviewer #2 (Remarks to the Author):

This revised manuscript responded to my concerns for the most part. I still have a negative reaction to their "uncharted" comment but they add a considerable amount of data in support of Hand2 contributing to the LPM and cells from the LPM that construct the mesothelia. Also added is a figure better explaining the embryonic contributions to mesothelial tumors. I particularly like the association with the PGCs and the progression of the mesothelium surrounding the yolk mass. As such the manuscript provides significant new information to an understanding of mesodermal development in zebrafish.

Reviewer #3 (Remarks to the Author):

The authors have spent an enormous effort to respond to the questions and reservations from all reviewers including mine. They share my view that the manuscript covers two issues related to developmental and disease aspects of which the first for most attention. My questions were focused on the disease aspects i.e. mesothelioma formation. The authors have addressed these questions and show a correlation between asbestos exposure and an increasing re-activation of early LPM factors active in mesothelial progenitors, in particular of Hand2. Causality however remains to be seen. Unfortunately the new data do not clearly indicate a relationship between tumor type, genetics and the expression of early LMP factors. However, this has been an open question and the data are what they are. I agree with the authors that the additional data provides strong support to investigate the role of re-activated LPM gene expression in mesothelial development and mesothelioma formation.

Reviewer #4 (Remarks to the Author):

The revised manuscript from Prummel et al. addresses the origins of the mesothelium in zebrafish. Using lineage tracing with transgenic markers of the lateral LPM, including *drl* and *hand2*, the authors show that *hand2*⁺ cells migrate laterally to form the parietal peritoneum and ventrally and medially to form the visceral peritoneum. The authors use single cell RNA-seq to define expression profiles of *hand2*⁺ cells that they think identify this progenitor population within the LPM. They provide evidence that there may be conservation for the *hand* transcription factors in the mesothelia via *cre*-mediated lineage tracing in mice. Examining zebrafish *hand2* mutants they find *hand2* is required for enclosure of the yolk by the parietal mesothelium and the internal organs by the visceral mesothelium, which the authors attribute to improper migration. Finally, they examine human mesotheliomas and show that some adopt a progenitor-like expression profile.

This study encompasses a significant amount of work. While predominantly descriptive, it provides new information about the development and expression of genes that give rise to these tissues in zebrafish and relates their findings to mesotheliomas in mouse models and humans. The descriptive aspects of the paper are overall well done. While the authors have done quite a bit to address the previous concerns there are some issues that could have been addressed better or were not brought up previously.

1. With respect to Reviewer #1's point about the authors' interpretations of improper migration being the primary causes of the peritoneal defects as opposed to cell specification and differentiation

defects, it does not appear that this point has been adequately addressed with the additional images and movie or in the text. The information provided in the manuscript (beginning with line 282), even with the new hand2:EGFP images in the hand2 mutants, does not rule out that the mesothelium progenitors are not specified or differentiate properly in the hand2 mutants. Based on the *drl:EGFP*, they do leave open that there could be a fate transformation, as pointed out previously. Furthermore, there are not images showing what these cells that are marked with the transgenic reporters look like at earlier stages in the mutants. If it were singularly a migration defect, would there not be a large build-up of cells in the LPM or close to the midline that fail to migrate? They also only discuss migration defects in hand2 mutants in the Results and Discussion (paragraph beginning on line 410). However, previous work from the Yelon lab has shown that hand2 is required for both the differentiation and migration of tissues, such as the heart. Minimally, unless the authors have evidence that can be included definitively showing the defects are only due to improper mesothelial progenitor migration, I suggest the authors need to discuss differentiation defects of the mesothelial progenitor cells in the hand2 mutants as an additional possibility left open, even if they favor a different hypothesis.

2. I am a bit surprised by how the PGC data are portrayed. Previous data from the Raz lab, who is an author, showed that hand2 mutants have PGC migration defects. However, I think these occur after the 18s stage (Weidinger et al, 2002). Subsequently, this group showed that SDF1/Cxcl12 deficient embryos also have PGC migration defects. It is not clear to me why the previous hand2 mutant experiments are not directly referenced and in the results explicitly included as part of the rationale, given the gene expression association from scRNA-seq made? It is also not clear why given the new tools available these effects on PGCs in the hand2 mutants were not revisited. Minimally, I think the connection with hand2 beyond the rationale from the scRNA-seq and migration defects in these mutants could be better articulated within the Results and Discussion.

Minor comments:

1. Sentence on lines 49-51 is awkward. "Contributions" and "contribute" in the same sentence.
2. Figure 6 would benefit from some additional labels in the panels (arrows and indication of where the sections are in the body – visceral or parietal peritoneum). Some of it is in the legend. It would be easier for the reader though if the panels were marked. There also appear to be faint GFP+ cells in the ventral portion of the lower panels for G and H.
3. A genetic photoconvertible method of lineage tracing with the hand2 line to mark the cells within the lpm and their migration paths would have nicely complemented and further make the cause about the different migration trajectories of the hand2+ cells.

REVIEWER COMMENTS

Reviewer #2 (Remarks to the Author):

This revised manuscript responded to my concerns for the most part. I still have a negative reaction to their “uncharted” comment but they add a considerable amount of data in support of Hand2 contributing to the LPM and cells from the LPM that construct the mesothelia. Also added is a figure better explaining the embryonic contributions to mesothelial tumors. I particularly like the association with the PGCs and the progression of the mesothelium surrounding the yolk mass. As such the manuscript provides significant new information to an understanding of mesodermal development in zebrafish.

We appreciate the reviewer's expert take on our revised manuscript. To amend the issue taken with the phrasing of “uncharted” mesothelium emergence, etc. we have now edited the paper's abstract to not state this term.

Reviewer #3 (Remarks to the Author):

The authors have spent an enormous effort to respond to the questions and reservations from all reviewers including mine. They share my view that the manuscript covers two issues related to developmental and disease aspects of which the first for most attention. My questions were focused on the disease aspects i.e. mesothelioma formation. The authors have addressed these questions and show a correlation between asbestos exposure and an increasing re-activation of early LPM factors active in mesothelial progenitors, in particular of Hand2. Causality however remains to be seen. Unfortunately the new data do not clearly indicate a relationship between tumor type, genetics and the expression of early LMP factors. However, this has been an open question and the data are what they are. I agree with the authors that the additional data provides strong support to investigate the role of re-activated LPM gene expression in mesothelial development and mesothelioma formation.

We thank the reviewer for their continued support of our work and input on the mesothelioma angle of our manuscript. We are keen to continue our work in this direction with the help of new genomics approaches that we hope will help linking re-activation of early LPM genes to particular types of mesothelioma tumors.

Reviewer #4 (Remarks to the Author):

*The revised manuscript from Prummel et al. addresses the origins of the mesothelium in zebrafish. Using lineage tracing with transgenic markers of the lateral LPM, including *drl* and *hand2*, the authors show that *hand2*⁺ cells migrate laterally to form the parietal peritoneum and ventrally and medially to form the visceral peritoneum. The authors use single cell RNA-seq to define expression profiles of *hand2*⁺ cells that they think identify this progenitor population within the LPM. They provide evidence that there may be conservation for the *hand* transcription factors in the mesothelia via *cre*-mediated lineage tracing in mice. Examining zebrafish *hand2* mutants they find *hand2* is required for enclosure of the yolk by the parietal mesothelium and the internal organs by the visceral mesothelium, which the authors attribute to improper migration. Finally, they examine human mesotheliomas and show that some adopt a progenitor-like expression profile.*

This study encompasses a significant amount of work. While predominantly descriptive, it provides new information about the development and expression of genes that give rise to these tissues in zebrafish and relates their findings to mesotheliomas in mouse models and humans. The descriptive aspects of the paper are overall well done. While the authors have done quite a bit to address the previous concerns there are some issues that could have been addressed better or were not brought up previously.

We thank the reviewer for their supportive take on our manuscript and our efforts to revise the manuscript according to the input of the initial three reviewers. In the revised manuscript, we retained the yellow highlighting of the changes introduced in the first revision, and marked changes introduced now as pink.

1. With respect to Reviewer #1's point about the authors' interpretations of improper migration being the primary causes of the peritoneal defects as opposed cell specification and differentiation defects, it does not appear that this point has been adequately addressed with the additional images and movie or in the text. The information provided in the manuscript (beginning with line 282), even with the new *hand2:EGFP* images in the *hand2* mutants, does not rule out that the mesothelium progenitors are not specified or differentiate properly in the *hand2* mutants. Based on the *drl:EGFP*, they do leave open that there could be a fate transformation, as pointed out previously. Furthermore, there are not images showing what these cells that are marked with the transgenic reporters look like at earlier stages in the mutants. If it were singularly a migration defect, would there not be a large build-up of cells in the LPM or close to the midline that fail to migrate? They also only discuss migration defects in *hand2* mutants in the Results and Discussion (paragraph beginning on line 410). However, previous work from the Yelon lab has shown that *hand2* is required for both the differentiation and migration of tissues, such as the heart. Minimally, unless the authors have evidence that can be included definitively showing the defects are only due to improper mesothelial progenitor migration, I suggest the authors need to discuss differentiation defects of the mesothelial progenitor cells in the *hand2* mutants as an additional possibility left open, even if they favor a different hypothesis.

Evidence for a fate transition of *hand2*-expressing cells in *hand2* mutants is indeed lacking, and previous work has mainly described migration and cell number changes in *hand2*-expressing cells (Yelon et al., 2000; Trinh et al., 2005; Schoenebeck et al., 2007, and more) and in adjacent tissues (i.e. Perens et al., 2016, yet deemed non-autonomous to the pronephric progenitors). Our lineage labeling-based characterization does indicate migration and proliferation defects, and our *in toto* imaging as added in the previous revision clearly shows disorganized *hand2:EGFP*-expressing cells at time points when wildtype embryos have formed beautiful mesothelial covering of the yolk (Figure 6). In our previous discussion, we thus linked our observations on migration defects with previous work in the field.

We have now revisited this phenotype using *hand2* morphants (using the main translation-blocking morpholino against *hand2* used in the field) and used Imaris plus manual corrections to quantify cells in wildtype versus *hand2*-perturbed embryos over the first 3 days of development. Morpholino injection enabled us to make sure we are imaging *hand2*-perturbed embryos. This analysis is now outlined in the two additional graph panels in Figure 6: Figure 6G shows the overall cell number trajectory in wildtype versus perturbation, which hints at a reduced cell number. However, as this analysis includes the highly dense *hand2:EGFP*-expressing heart and pectoral fin fields, we also performed isolated cell counts on the trunk area caudal of the pectoral fin to more specifically focus on the mesothelial phenotype. We added example images from the light sheet datasets used for the quantification as SFig 8 for completion. This analysis clearly reveals that *hand2* perturbation causes a reduction in overall *hand2:EGFP* cell number. Together with the failed mesothelial migration, reduced cell number is in line with the previous reports pointed out by the reviewer (which we also reference in our manuscript). We are keen to follow this phenotype impact further in our future work, and have now added the improved phenotype description to the main text. We have also amended the discussion according to the reviewer's input.

While by no means definitive (as we also indicate in the manuscript text), we believe our combined data provides a new angle in the interpretation of the *hand2* loss-of-function phenotype in zebrafish and possibly also other vertebrate models.

2. I am a bit surprised by how the PGC data are portrayed. Previous data from the Raz lab, who is an author, showed that *hand2* mutants have PGC migration defects. However, I think these occur after the 18s stage (Weidinger et al, 2002). Subsequently, this group showed that *SDF1/Cxcl12* deficient embryos also have PGC migration defects. It is not clear to me why the previous *hand2* mutant experiments are not directly referenced and in the results explicitly included as part of the rationale, given the gene expression association from *scrNA-seq* made? It is also not clear why given the new tools available these effects on PGCs in the *hand2* mutants were not revisited. Minimally, I think the connection with *hand2* beyond the rationale from the *scrNA-seq* and migration defects in these mutants could be better articulated within the Results and Discussion.

The reviewer hits upon an important aspect of our overall characterization of mesothelial progenitors. Based on the reviewer's input, we have now added context to the Results part and to the Discussion paragraph referencing the PGC experiments. Pandemic restrictions and circumstances have kept us from pursuing the connection of SDF1/Cxcl12 and Hand2 any further but we are keen to do so!

Minor comments:

1. Sentence on lines 49-51 is awkward. "Contributions" and "contribute" in the same sentence.

Great catch – we have now amended the sentence to read:

"Studies across vertebrate models have revealed key insights into **the role** of Hand2 and its paralog eHand/Hand1 in anterior LPM (ALPM) progenitors that contribute to the heart, (fore)limbs, and branchial arches[...]"

2. Figure 6 would benefit from some additional labels in the panels (arrows and indication of where the sections are in the body – visceral or parietal peritoneum). Some of it is in the legend. It would be easier for the reader though if the panels were marked. There also appear to be faint GFP+ cells in the ventral portion of the lower panels for G and H.

We had now amended additional labels the Figure 6. Abbreviations to point out different organs and arrows to indicate visceral and parietal peritoneum / body wall. We agree with the reviewer that this eases the interpretation of the figure.

3. A genetic photoconvertible method of lineage tracing with the hand2 line to mark the cells within the lpm and their migration paths would have nicely complemented and further make the cause about the different migration trajectories of the hand2+ cells.

We absolutely agree with the reviewer on this point. While our live imaging over several days does provide an overview of the distinct and dispersed migration paths of *hand2*-expressing mesothelial progenitors, direct lineage-labeling of mesothelial progenitors remains a highly desirable experimental direction.

We have spent significant effort into establishing Dendra2-based photoconversion tools (i.e. Felker et al., 2018), which we have previously used to map cardiac lineages. A major obstacle towards performing such experiments for the *hand2*-marked mesothelial progenitors remains the challenging generation of *hand2*-based transgenic tools: the regulatory elements for the early expression (i.e. as critical to our work) of *hand2* remain unknown; the only transgenic recapitulating the early LPM expression of *hand2* remains the beautiful BAC-based *hand2:EGFP* line as also used in our work and for our quantifications outlined above.

Transgenics based on *drl* that label the entire emerging LPM are not active for long enough and strong enough to follow Dendra2 expression outside the heart field and hematopoietic lineages beyond 24 hpf (see also Prummel et al., 2019), and our other transgenic lines such as *tbx1:Dendra2* (Felker et al., 2018) do not label all relevant *hand2*-expressing cells early on. In our future work, we are looking forward to using knockin-based *hand2* transgenics, which are currently being generated.

REVIEWERS' COMMENTS

Reviewer #4 (Remarks to the Author):

The second revision of the manuscript from Prummel et al. addresses the origins of the mesothelium in zebrafish. Furthermore, they provide evidence for conservation of the hand transcription factors in the mesothelia via cre-mediated lineage tracing in mice, as well as examine gene expression profiles human mesotheliomas. The authors have addressed the major concerns over the interpretation of migration vs differentiation and the background of the PGC data. How they have performed the additional analysis and incorporated the data are fine. Their interpretations are consistent with the new data. The manuscript provides new understanding of the origins and development of the mesothelium in vertebrates and potentially insights into diseases affecting this tissue.

REVIEWERS' COMMENTS

Reviewer #4 (Remarks to the Author):

The second revision of the manuscript from Prummel at al. addresses the origins of the mesothelium in zebrafish. Furthermore, they provide evidence for conservation of the hand transcription factors in the mesothelia via cre-mediated lineage tracing in mice, as well as examine gene expression profiles human mesotheliomas. The authors have addressed the major concerns over the interpretation of migration vs differentiation and the background of the PGC data. How they have performed the additional analysis and incorporated the data are fine. Their interpretations are consistent with the new data. The manuscript provides new understanding of the origins and development of the mesothelium in vertebrates and potentially insights into diseases affecting this tissue.

We much appreciate the reviewer's time and efforts to assess the finalizing version of our manuscript. The reviewer's input has helped to further consolidate also the other reviewers' previous comments to improve the packaging and messaging of our manuscript. We are looking much forward to continuing our work on mesothelial biology using the foundation laid in our work presented here.